# Ideas and perspectives: Emerging contours of a dynamic exogenous kerogen cycle

Thomas M. Blattmann[1*]

[1]Biogeochemistry Research Center, Research Institute for Marine Resources Utilization, Japan Agency for Marine-Earth
Science and Technology (JAMSTEC), 2-15 Natsushima-cho, 237-0061 Yokosuka, Japan
[*]Present address: Geological Institute, ETH Zurich, Sonneggstrasse 5, 8092 Zurich, Switzerland

*Correspondence to*: Thomas M. Blattmann (blattmannt@jamstec.go.jp)

**Abstract.** Growing evidence points to the dynamic role that kerogen is playing on Earth's surface in controlling atmospheric chemistry over geologic time. Although quantitative constraints on the weathering of kerogen remain loose, its changing weathering behavior modulated by the activity of glaciers suggests that this largest pool of reduced carbon on Earth may have played a key part in atmospheric $CO_2$ variability across recent glacial-interglacial cycles and beyond. This work enunciates the possibility of kerogen oxidation as a major driver of atmospheric $CO_2$ increase in the wake of glacial episodes. This hypothesis of centennial and millennial-timescale relevance for this chemical weathering pathway is substantiated by several lines of independent evidence synthesized in this contribution including the timing of atmospheric $CO_2$ increase, atmospheric $CO_2$ isotope composition ($^{13}C$ and $^{14}C$), kerogen oxidation kinetics, observations of kerogen reburial, and modeling results. The author hypothesizes that the deglaciation of kerogen-rich lithologies in western Canada contributed to the characteristic deglacial increase in atmospheric $CO_2$, which reached an inflection point $\leq300$ years after the Laurentide Ice Sheet retreated into the kerogen-poor Canadian Shield. To reconcile the release of isotopically light carbon via kerogen oxidation with Earth surface carbon pool constraints, major oceanic degassing and biospheric regrowth must have acted in concert across glacial-interglacial transitions. Additionally, a process such as a strong shift in the ratio of $C_3$ to $C_4$-derived organic matter must be invoked to maintain isotope mass balance, a point key for reconciling the hypothesis with the carbon isotope record of marine dissolved inorganic carbon. In order to test this hypothesis, quantitative constraints on the contribution of kerogen oxidation to $CO_2$ rise at glacial terminations are needed through systematic studies on (1) $CO_2$ fluxes emanating from the weathering of different lithologies, (2) oxidation kinetics of kerogen along glacial chronosequences, and (3) high-resolution temporal changes in the aerial extent of glacially exposed lithological units and glacial flour.

## 1 Introduction

Over geologic timescales, atmospheric $CO_2$ is controlled by the combined effects of chemical weathering of silicates and carbonates and the organic carbon cycle (Berner et al., 1983; Berner, 1990; Torres et al., 2014). Organic carbon in the form of

kerogen[1] comprises around 15 million PgC, which encompasses over 99.9% of reduced carbon present on Earth. Tectonic
uplift and denudation subjects 150 PgC/kyr of kerogen to weathering on Earth's surface (Hedges and Oades, 1997) thereby
facilitating entry of this geologically ancient carbon into the atmosphere. Upon oxidation of kerogen, $O_2$ is consumed and $CO_2$
is released to the atmosphere. In reverse, biospheric organic carbon burial in marine sediments removes carbon from Earth's
surface thereby drawing down atmospheric $CO_2$ and increasing $O_2$ over geological timescales (Galy et al., 2008). Therefore,
kerogen weathering and sedimentary organic matter burial in the ocean play compensatory roles in governing atmospheric
chemistry (Fig. 1) with kerogen oxidation considered important for atmospheric chemistry over million-year timescales (e.g.,
Petsch, 2014; Bolton et al., 2006).

However, the decay of kerogen on Earth's surface is incomplete (e.g., Hemingway et al., 2018; Leythaeuser, 1973), with
physical erosion followed by riverine transport (Galy et al., 2015) and reburial in lacustrine and marine settings (e.g., Blattmann
et al., 2018a; Blattmann et al., 2019b; Cui et al., 2016; Sackett et al., 1974; Sparkes et al., 2020). The operation of this "simple
cycle besides the more complicated common circulation of carbon", enunciated by Sauramo (1938), begs the questions (i)
what is the reburial efficiency of kerogen, (ii) what is the weathering efficiency of kerogen, (iii) what are their controlling
factors, and (iv) how do reburial and weathering efficiency vary over geologic time and space? The answers presented will
suggest that the exogeneous kerogen cycle behaves dynamically with this contribution hypothesizing a connection between
kerogen oxidation and millennial/centennial-scale atmospheric $CO_2$ increases during glacial terminations.

## 2 Carbon isotopes and contradictions?

Driven mainly by orbital cyclicity, glaciers have rhythmically waxed and waned across Earth's surface together with changes
in atmospheric chemistry, including the greenhouse gas $CO_2$ (Hays et al., 1976: Barnola et al., 1987). The amount and carbon
isotope composition of atmospheric $CO_2$ (Fig. 2) depend on (1) the terrestrial biospheric organic carbon pool, (2) the dissolved
inorganic carbon (DIC) pools residing in the surface and deep ocean, (3) exchanges between the atmosphere and the terrestrial
biosphere and ocean, and (4) the export of organic matter and carbonate from the surface to deep waters and sediments of the
oceans (Sigman and Boyle, 2000). During deglaciation, an increasingly large terrestrial biospheric carbon pool (Bird et al.,
1994; Shackleton, 1977; Shackleton et al., 1983) is inferred to have controlled an increase in the stable carbon isotope
composition of marine DIC, reconstructed primarily from foraminifera (e.g., Schmittner et al., 2017). However, during times
of most rapid $CO_2$ increase during transitions from glacial to interglacial periods, negative stable carbon isotope shifts in
atmospheric $CO_2$ occurred (Fig. 3; Smith et al., 1999; Schmitt et al., 2012). This is a strong indicator that, in conjunction with
ocean-air exchange, respired organic matter acted as a direct source of carbon to the atmosphere (Bauska et al., 2016). Organic
matter can be categorized into biospheric and lithogenic (i.e., kerogen) forms, with both exhibiting similar [13]C signatures yet
modern and dead [14]C signatures, respectively (e.g., Lewan, 1986; Meyers, 1994). The constraints imposed by radiocarbon

---

[1] In this work, kerogen is used as an umbrella term for all rock-derived forms of reduced carbon including soluble, insoluble,
rock disseminated, rock forming, solid and liquid forms, as well as fossil palynomorphs, biogenic and abiogenic graphite.

indicate there was a source of carbon to the atmosphere and ocean depleted in or devoid of radiocarbon (Broecker and Clark 2010; Hain et al., 2014; Rafter et al., 2019; Zhao et al., 2018), thereby limiting potential contributions from modern biospheric organic carbon sources. Other studies have proposed that carbon sourced from deep ocean DIC was the predominant source for carbon transferred to the atmosphere during glacial terminations (e.g., Hain et al., 2014). However, this hypothesis appears inconsistent with the negative fluctuation observed in the $^{13}C$ fingerprint of atmospheric $CO_2$ (see discussion in Broecker and McGee, 2013). This leaves us with some apparent contradictions. If there was a source of oxidized organic matter to the atmosphere, how is this reconcilable with the changes in the carbon isotope trajectory of marine DIC? From the author's perspective, this is where a new idea is needed.

Unlike the global, nearly unison rhythm of the glacial-interglacial marine oxygen isotope record, the global deglacial increase in carbon isotopes shows a notable exception: For much of the North Atlantic, the Holocene stable carbon isotope values of DIC are lighter than those of the Last Glacial Maximum (e.g., Peterson et al., 2020; Broecker and McGee, 2013; Bouttes et al., 2020). From the author's perspective, this is notable because the northernmost Atlantic is the locus of major downwelling driving global thermohaline circulation (de Carvalho Ferreira and Kerr, 2017). The negative shift observed in the North Atlantic was modeled by Crichton et al. (2016) showing this is explainable by the marine uptake and subduction of light carbon released to the atmosphere by terrestrial organic matter oxidation (hypothesized as permafrost in their case). Furthermore, Crichton et al. (2016) model the positive carbon isotope shift in DIC observed in the South Atlantic demonstrating that contemporaneous positive and negative shifts in different ocean sectors are reconcilable with the release of isotopically light carbon from land. On a global scale, how could the stable carbon isotope mass balance add up if a scenario of organic matter oxidation is assumed? In addition, is it reconcilable with the global marine DIC carbon isotope shift? An example of such a budget is shown in Table 1, which pegs atmospheric $CO_2$ and marine DIC stable carbon isotope shifts to recorded values and uses constraints of carbon pools sizes (see table caption for references). While the constraints lead to an array of solutions depending on the degrees of freedom chosen and which literature values one accepts for the various parameters, a plausible budget is attempted here. Based on this scenario of 600 PgC released to the atmosphere by kerogen oxidation and 520 PgC degassed from the oceans, isotope mass balance constraints suggest a glacial-interglacial regrowth of the biosphere on the order of 920 PgC. To maintain isotope mass balance, this solution assumes a constant marine dissolved OC pool size and requires a strong increase in the ratio of $C_3$ to $C_4$ vegetation-derived biomass from approximately 1:1 until the ratio reaches the modern-day value of about 4:1 (Still et al., 2003), which is plausible given the large-scale change in vegetation across glacial-interglacial cycles (Adams et al., 1990; Huang et al., 2001). The terrestrial OC pool increase of 920 PgC is larger than usually computed using carbon isotopes of marine DIC, but lies in the ballpark expected from palynological observations (see Table 1 in Zeng, 2003; see also Crowley, 1995 and discussion in Jeltsch-Thömmes et al., 2019). These calculations are back-of-the-envelope as constraints like the glacial-interglacial change in the carbon isotope composition of DIC carry heavy weight and considerable uncertainty (+0.34±0.19‰ 2-σ, Peterson et al., 2014). Additionally, the fitted values $C_4$ and $C_3$ vegetation amounts are sensitive to other parameters such as the change in marine DIC pool size and are therefore limited in their value to proof of concept: These calculations show that the release of $CO_2$ from organic matter such as kerogen is compatible with

1) the negative perturbation in atmospheric $CO_2$ during the glacial-interglacial transition, 2) the increase in the carbon isotope composition of global marine DIC, and 3) general carbon pool size constraints. Complementing the widely supported hypotheses of an increasing terrestrial biospheric organic carbon pool and major changes in ocean-atmosphere $CO_2$ exchange (e.g., Shackleton, 1977; Lindgren et al., 2018; Sigman and Boyle, 2000), the author will propose major release of $CO_2$ to the atmosphere via kerogen oxidation during deglaciation.

## 3 Kerogen and glaciers – Dynamic modulators of the global carbon cycle?

Chemical weathering of silicates and carbonates as well as the burial and oxidation of organic matter exert fundamental control over atmospheric chemistry on Earth over geologic timescales (Berner et al., 1983; Berner, 1990: Hilton and West, 2020; Petsch, 2014). On modern-day Earth, carbon sources and sinks from mineral weathering (i.e., silicates and carbonates via carbonic and sulfuric acids) and biogeochemical fluxes (i.e., organic matter burial and kerogen oxidation) stand in fine balance to one another with contrasting concoctions of these under different tectonosedimentary settings (e.g., Bufe et al., 2021; Hilton and West, 2020; Blattmann et al., 2019a; Horan et al., 2019). Deglaciation likely modulated silicate and carbonate weathering resulting in enhanced (temporary) drawdown of atmospheric $CO_2$ (e.g., Tranter, 1996; Gibbs and Kump, 1996; Munhoven and François 1996; c.f., Schachtman et al., 2019). Glaciers have been invoked as agents for accelerating chemical weathering of carbonate and silicate minerals by increasing sediment yield and creating high surface area, reactive substrate (Torres et al., 2017; Vance et al., 2009; Yu et al., 2021), with carbonate weathering constituting a source of $CO_2$ to the atmosphere when sulfuric acid, stemming from oxidized sulfide minerals, is involved (c.f., Torres et al., 2014; Kölling et al., 2019). In the following, the idea of glaciers producing reactive substrate susceptible to chemical weathering is reviewed and explored for kerogen.

Evidence accumulated from over a century of scientific studies supports the idea that the reburial of kerogen was more extensive during cold interludes in Earth history where glacial erosion and ice rafting was widespread (Blattmann et al., 2018b). This general pattern of kerogen reburial is observed spatially today with more efficient reburial in high latitude glaciated regions than in low latitude sedimentary systems (Cui et al., 2016). This enhanced reburial efficiency strengthens the short-circuiting of the exogenous kerogen cycle by keeping this ancient, reduced carbon locked away. However, under glaciofluvial conditions, the entry of kerogen-bound carbon into surficial carbon pools is promoted (Horan, 2018), with glacial meltwater releasing dissolved organic matter from kerogen, which is readily degraded and consumed by microbes (Hood et al., 2009; Schillawski and Petsch, 2008; Sharp et al., 1999). Additionally, frost shattering, together with the retreat of glaciers, exposes finely ground, high surface area, reactive sediment, thereby accelerating oxidation and the release of kerogen-derived $CO_2$ to the atmosphere (Fischer and Gaupp, 2005; Horan et al., 2017). The initially strong input of kerogen-derived $CO_2$ would die down as the availability of glacially ground, reactive kerogen declines into an interglacial period. Enhanced degradation of kerogen in the wake of glacial episodes is consistent with observations from areas of ongoing deglaciation (Horan et al., 2017). Contemporaneous perturbations in $CO_2$ fluxes are expected to result for silicate and carbonate weathering via carbonic and

sulfuric acids (Vance et al., 2009; Torres et al., 2014). However, kerogen oxidation, owing to its faster weathering kinetics and direct conversion to $CO_2$ leads to considerable losses in kerogen content over decadal timescales (Horan et al., 2017 and references therein; Fischer et al., 2007). This is a process by which $CO_2$ can be injected directly into the atmosphere and impact glacial-interglacial cycles (Fig. 2).

    In the following, quantitative arguments are presented: Of the approximately 150 PgC/kyr of kerogen reaching the Earth's

surface ($F_L$; Hedges and Oades, 1997), $43^{+61}_{-25} \frac{PgC}{km^2 yr}$ is currently exported by rivers to oceans ($F_{L-O}$; Galy et al., 2015; Copard et al., 2007), indicating that the modern-day reburial efficiency of this carbon lies in the ballpark of 30% (10-70%). Normalized to global land area ($A_L = 149,000,000 \ km^2$), the flux of carbon due to oxidation of kerogen to the atmosphere ($F_{L-A}$) can be calculated from:

$F_{L-A} = \frac{F_L - F_{L-O}}{A_L}$           (Equation 1)

    Here, equation 1 yields a modern-day global $F_{L-A}$ of $0.72^{+0.17}_{-0.41} \frac{MgC}{km^2 yr}$. Reports of local and regional $CO_2$ fluxes from kerogen oxidation are based on disparate study sites and methods (Table 2). From this, reported $CO_2$ fluxes stemming from oxidation of rock disseminated forms of kerogen and across larger catchment areas under aerobic conditions span over two orders of

magnitude ranging from 0.3 to 64 $\frac{MgC}{km^2 yr}$. The only quantities available for catchments of ongoing deglaciation stem from the Southern Alps of New Zealand, which are one to two orders of magnitude faster than those of the average Earth surface (ranging between 9-50 $\frac{MgC}{km^2 yr}$; Horan et al., 2017). Taking this flux range as a working estimate, how would kerogen oxidation translate into changes in atmospheric $CO_2$ concentration? In Zeng's (2003) modeling work postulating soil oxidation upon retreat of overriding glaciers, the effect of continuous inputs of organic matter-derived carbon directly to the atmosphere are

calculated: 550 PgC oxidized lead to 60 PgC increase in atmospheric $CO_2$ (30 ppm increase). Similarly, modeling by Simmons et al. (2016) also propose the oxidation of overridden soil organic carbon subsequent to deglaciation and calculate a 600 PgC release resulting in a 40-60 ppm increase in atmospheric $CO_2$ (c.f., also Ciais et al. (2012) who calculate the deglacial oxidation of 700 PgC of an "inert" terrestrial organic carbon pool). The idea of soil oxidation is mathematically analogous to kerogen oxidation in the hypothesis presented here. How much area of deglaciated terrain ($A_D$) would therefore be needed to release

550-600 PgC of kerogen-derived carbon? Using Zeng's (2003) and Simmons et al.'s (2016) carbon release estimates ($550 \ PgC \le C_R \le 600 \ PgC$) and Horan et al.'s (2017) kerogen oxidation kinetics ($9 \frac{MgC}{km^2 yr} \le F_{KO} \le 50 \frac{MgC}{km^2 yr}$) over a 6,000-year time window ($T_{G-I}$, timespan of pronounced $CO_2$ increase over the glacial-interglacial transition; see Fig. 3) and subtracting global average baseline kerogen oxidation ($F_{L-A} \approx 0.72^{+0.17}_{-0.41} \frac{MgC}{km^2 yr}$; see Eq. 1):

$A_D = \frac{C_R}{T_{G-I} * (F_{KO} - F_{L-A})}$           (Equation 2)

With calculated $A_D$ yielding between 2 and 12 million km$^2$, in the most conservative estimate, 12 million km$^2$ of deglaciated terrain containing sedimentary and metasedimentary rocks and their debris (glacial flour accumulated over millennia of glaciation) would cover the source. In comparison, Canada covers approximately 10 million km$^2$, which was mostly covered by the Laurentide Ice Sheet during the Last Glacial Maximum. With this, a plausible scenario for releasing kerogen-derived $CO_2$ to the atmosphere that could account for a 30-60 ppm rise during the glacial-interglacial transition encompassing an area equal to or less than the terrestrial extent of the Laurentide Ice Sheet is identified. Similar to the oxidation fluxes reported from deglaciating catchments (Horan et al., 2017) are those reported for the in-situ weathering of massive outcrop formations of marls and shales (e.g., Soulet et al., 2018; Littke et al., 1991; see Table 2). Kerogen oxidation fluxes appear higher for catchments showing larger glacial coverage (Horan et al., 2017) suggesting that the relative contribution of freshly exposed and ground bedrock is highest for these areas and that kerogen oxidation fluxes progress over the course of deglaciation. The $CO_2$ flux emanating from kerogen oxidation in soils formed from glacial till (Keller and Bacon, 1998) suggests that oxidation fluxes an order of magnitude greater than the global average can sustain for millennia after deglaciation. However, a systematic chronosequence understanding is currently lacking that allows conclusions to be drawn as a variety of factors (e.g., lithology, initial kerogen content, grain size) exert control over kerogen oxidation (c.f., Fischer and Gaupp, 2005; Bolton et al., 2006; Martínez and Escobar, 1995). The bedrock and river sediments of the Southern Alps are relatively lean in kerogen (Copard et al., 2007; Horan et al., 2017). Therefore, for deglaciating landscapes exposing glacially ground kerogen-rich lithologies, larger oxidation fluxes than those reported from the Southern Alps can be expected, increasing and decreasing the plausible ranges for $F_{KO}$ and $A_D$, respectively (see next section).

Long-term accelerated decline in the radiocarbon concentration of atmospheric $CO_2$ parallel to an overall increase in $CO_2$ amount occurred since the Last Glacial Maximum (Reimer et al., 2013; Roth and Joos, 2013). Consistent with such a dilutional process, kerogen oxidation releases radiocarbon-dead $CO_2$ to the atmosphere. During the time period of greatest $CO_2$ increase from 17.5 until 11.5 kyr before present, the rate of decline in the concentration of $^{14}C$ in atmospheric $CO_2$ is greatest and averages about 35 ‰/kyr (Fig. 3; c.f., Broecker and Barker (2007)). With the release of 500 PgC (radiocarbon-dead) to the atmosphere over deglaciation, a drop in 100 to 200 ‰ $\Delta^{14}C$ of atmospheric $CO_2$ can be expected (see calculations by Zeng, 2007). Kerogen oxidation would also release isotopically light carbon to the atmosphere, which is also consistent with the atmospheric record (Fig. 3; Bauska et al., 2016). Finally, driven by orbital forcing (Hays et al., 1976), the response in atmospheric $CO_2$ faithfully echoing increasing global temperatures and diminishing glaciated terrain (Sigman and Boyle, 2000; Stips et al., 2016) goes hand in hand with continuous aerial exposure and enhanced oxidation of finely ground, reactive kerogen (Fig. 2).

## 4 A Canadian tale. And what about microbes?

Kerogen exposed on Earth's surface is distributed unevenly, with kerogen-rich surface lithologies extending across much of western Canada (see Fig. 2 in Copard et al., 2007). These areas experienced dramatic deglaciation across the 17.5-11.5 ka timeframe (Fig. 4; Dyke, 2004; Dalton et al., 2020). In addition to the widespread occurrence and high abundance of rock disseminated forms of kerogen, there are also notable surficial occurrences of hydrocarbon-rich lithologies including coal and oil sands within the deglaciated terrains of western Canada. Within the Province of Alberta, Andriashek and Pawlowicz (2002) report the widespread occurrence of reworked shale and bitumen in Quaternary till stemming from Cretaceous shales and the oil sands, with unoxidized and oxidized forms present, and the latter enhanced by aerial exposure across paleosurfaces. Today, bituminous erratic boulders are found strewn around the region over tens of thousands of square kilometers (Rutherford, 1928; Andriashek, 2018). As Andriashek (2018) points out, Rutherford (1928) states: "There are perhaps many more occurrences of bituminous sand within the glacial deposits of Alberta, but since they weather comparatively readily and become covered with soil, they are not likely to be detected unless by accident, …". This observation of rapid weathering of bitumen indicates that today's occurrences in glacial tills represent the tip of the iceberg of what once was present. Laboratory incubations simulating $CO_2$ respiration from bituminous materials reveal fluxes that are markedly higher than those associated with oxidation of rock disseminated forms of kerogen (see Table 2). Chang and Berner (1998, 1999) report subaquatic bituminous coal oxidation (calculated $>300 \frac{MgC}{km^2 yr}$) releasing $CO_2$ at rates 1-2 orders of magnitude higher than those reported for rock disseminated kerogen, and 3 orders of magnitude greater than the average for Earth's surface. Microbes likely play key roles in assimilating and releasing this ancient organic carbon as $CO_2$ to the atmosphere (ZoBell, 1946; Hemingway et al., 2018) with microbial communities on the surface colonizing oil sand-derived bitumen under both summer and winter conditions (Wyndham and Costerton, 1981; Wong et al., 2015). Biodegradation is accelerated during summer months when temperatures of subaerially exposed outcrops of oil sands reach 60°C (Wong et al., 2015). Field experiments show a 2.2-fold increase in kerogen-derived $CO_2$ with 10°C increase (Soulet et al., 2021). Microbial degradation experiments on bitumen (Ait-Langomazino et al., 1991) reveal an even greater $CO_2$ release when extrapolated to natural systems (e.g., oil sands, see Table 2) exceeding $18 \frac{MgC}{km^2 day}$, over one hundred times greater than the highest $CO_2$ flux released by rock disseminated kerogen oxidation. Other laboratory-based studies (e.g., Uribe-Alvarez et al., 2011) that investigated the oxidative decay of hydrocarbon fractions also suggest similarly high fluxes when scaled to natural systems, even though these studies were conducted over periods of only a few weeks. Such observations likely overestimate long-term natural fluxes over multiple years and decades, while the fluxes reported by Chang and Berner (1998, 1999) likely represent an underestimate due to sample storage for several years prior to analysis and the absence of microbial activity under their experimental conditions. Additionally, comparable quantities of kerogen-derived $CO_2$ can also be released under anaerobic conditions, which may become relevant under warming glaciers (e.g., Bertassoli Jr. et al., 2016; Rogieri Pelissari et al., 2021; see also Sharp et al., 1999; Sharp and Tranter, 2017). Overall, it is conceivable that super carbon source terrains (hypothetically, areas laced or covered with coal, bituminous materials, etc.)

across western Canada could supply an overproportionate quantity of radiocarbon dead $CO_2$ to the atmosphere during glacial-interglacial transitions.

By 11.5 ka before present, the pronounced rise in atmospheric $CO_2$ had subsided. By this time, the Laurentide Ice Sheet had retreated beyond the easternmost reaches of the Western Canadian Sedimentary Basin and was exposing bedrock from the relatively kerogen-poor, highly metamorphosed Canadian Shield (Fig. 4; Dalton et al., 2020). The timing of the inflection point in the rise of $CO_2$ occurs ≤300 years after the entire western edge of the Laurentide Ice Sheet advanced onto the Canadian Shield suggests diminishing decay of kerogen and relaxation of the landscape. By this time, the rate of soil organic carbon

sequestration in these paraglacial landscapes begins to pick up substantially (Harden et al., 1992). Lithologies of the Fennoscandian Shield and most of its adjacent landmasses (deglaciated eastern and southern sectors of the Fennoscandian Ice Sheet; see Stroeven et al., 2016 and Copard et al., 2007) contain relatively low kerogen contents, therefore limiting its contribution to atmospheric $CO_2$ rise in the wake of Fennoscandian Ice Sheet retreat. Glacially ground graphite shed from these highly metamorphosed rocks is chemically recalcitrant and is mostly redeposited (Sauramo, 1938; Sackett et al., 1974;

Sparkes et al., 2020). Geophysical arguments propose that the relatively narrow timing of the Laurentide Ice Sheet's isochrons retreating into the Canadian Shield is more than coincidence: the lithological base and its overlying glacial till, influence the dynamics of ice sheet retreat with the Western Canadian Sedimentary Basin representing a deformable base while the Canadian Shield behaves rigidly providing an explanation for why the geological and ice sheet isochron map show similarities (Clark, 1994 and see Fig. 3 therein; Licciardi et al., 1998). At the same time in the context of the kerogen oxidation hypothesis, this

offers an explanation for the well-defined inflection point in the rise of atmospheric $CO_2$ characteristic for glacial-interglacial transitions.

In the bigger picture, during Earth's recent glacial episodes, the Laurentide Ice Sheet was the most extensive element of the cryosphere that waxed and waned across the continents (Batchelor et al., 2019) and, in conjunction with its lithological underpinning (Copard et al., 2007), likely played the largest role in releasing kerogen-derived $CO_2$ to the atmosphere upon

glacial retreat. However, estimates of $CO_2$ fluxes emanating from bedrock show a wide range (Table 2) and there is considerable uncertainty in our current state of knowledge: Dedicated biogeochemical weathering studies that provide estimates of $CO_2$ fluxes from bedrock-derived kerogen under relevant environmental conditions and over appropriate timescales are lacking. Chronosequence studies of kerogen oxidation rates in deglaciated terrains are needed to provide constraints on time-integrated $CO_2$ release to the atmosphere. In tandem with this, quantification is needed for (temporary)

kerogen reburial in subaerial and subaquatic terrestrial systems (e.g., moraines, lakes) on global and regional scales (e.g., Meybeck, 1993; Vonk et al., 2016; Blattmann et al., 2019b; Fox et al., 2020). Such information in conjunction with high resolution reconstructions of changes in land ice extent and the lithologies of bedrock and glacial till (including data on kerogen content and its stable carbon isotope composition) exposed by glacial retreat can, in theory, quantitatively disentangle the time-integrated contribution of kerogen-derived $CO_2$ to the atmosphere during glacial-interglacial transitions.

## 5 Tackling geologic deep time

While modern day kerogen weathering and reburial efficiencies are only loosely constrained, even less is known about how they varied back in geologic time. Overarching controls on these processes include the mode of erosion and transport ranging from glacial to glaciofluvial to fluvial, remineralization intensity, controlled by continental margin type and geomorphology, and the intrinsic reactivity of the kerogen present locally (Blair and Aller, 2012; Blattmann et al., 2018b). Also important are the bedrock lithology and regolith composition which have been hypothesized to exert feedback on temporal patterns of glacial-interglacial cyclicity (e.g., Roy et al., 2004; Zeng, 2007). For the Paleocene-Eocene Thermal Maximum, an extreme greenhouse episode, evidence increasingly suggests that enhanced remineralization and leaching of kerogen (Boucsein and Stein, 2009; Lyons et al., 2019), possibly enhanced by the activity of microbes (Hemingway et al., 2018; Petsch et al., 2001), increased the flux of carbon entering actively circulating pools on Earth's surface. Over Earth's history on $10^9$ year timescales, the reburial efficiency of kerogen presumably varied as a function of atmospheric $O_2$ content, with lower $O_2$ contents tied to higher reburial efficiency (Daines et al., 2017). Kerogen is surmised to have acted as a major source of carbon to the atmosphere and as an "antioxidant" during the early rise of atmospheric $O_2$ (Daines et al., 2017; Kump et al., 2011) and balancing $O_2$ levels over the Cenozoic (Galvez, 2020; see also Derry and France-Lanord (1996) for discussion on Paleogene). Across multiple glacial-interglacial cycles, enhanced kerogen oxidation would also be consistent with declining atmospheric $O_2$ on $10^6$ yr timescales (Stolper et al., 2016).

In order to understand changes in atmospheric chemistry through geologic time, in addition to comprehensively budgeting the effect of mineral chemical weathering (Blattmann et al., 2019a; Horan et al., 2019; Hilton et al., 2014), the changing efficiency of the oxidation and reburial of kerogen needs to be evaluated. Direct quantification of kerogen found reburied in sediments is often associated with considerable uncertainty owing to uncertainties in organic matter source apportionment (e.g., Lin et al., 2020; Blattmann et al., 2019b) and uneven spatial dispersal (e.g., Berg et al., 2021; Blattmann et al., 2018a; Cui et al., 2016). While radiocarbon was paramount for quantifying and establishing the importance of the reburial of kerogen in recent times (Blattmann et al., 2018b), its utility diminishes quickly for strata that pre-date the Last Glacial Maximum owing to its radioactive decay. However, associated with kerogen are a promising suite of trace elements and their respective isotope signatures including, among others, rhenium (Hilton et al., 2014; Horan et al., 2017), osmium (Georg et al., 2013; Ravizza and Esser, 1993), and iodine (Moran et al., 1998) as well as clay minerals (Blattmann and Liu, 2021), that can be exploited to trace sedimentary kerogen and its degradation. In the case of osmium, seawater records reveal isotopic shifts at the beginning of interglacial periods that are attributable to kerogen oxidation (Georg et al., 2013; compare discussions in Peucker-Ehrenbrink and Ravizza, 2020), consistent with the hypothesis presented here. More research constraining the exogenous kerogen cycle by quantification of reburied kerogen inputs (e.g., iodine isotopes, organic petrology) and kerogen oxidation recorded by chemical weathering proxies (e.g., osmium isotopes) is needed to put the presented hypothesis to the test.

## 6 Synthesis and outlook

In a nutshell, the hypothesis presented proposes the following: Less than 300 years after the Laurentide Ice Sheet retreated east past the easternmost edges of the kerogen-rich Western Canadian Sedimentary Basin, atmospheric $CO_2$ levels stabilized and the rate of decrease in $^{14}C$ concentrations of $CO_2$ subsided (Figs. 3 and 4). This inflection point is mirrored in the lithologies of the Canadian Shield that were exposed during deglaciation, which contain relatively minor amounts of reactive kerogen. Within the context of the presented hypothesis, the coincidence in time of global trends in atmospheric chemistry with spatiotemporal patterns in the distribution of freshly exposed deglaciated terrain impregnated with oxidizable and biodegradable kerogen and bituminous materials suggests that a burst (or bursts) of respired $CO_2$ contributed to the characteristic deglacial increase in atmospheric $CO_2$. As soon as glacially ground shales and bituminous materials either exhausted their labile kerogen content and/or became buried by soil and vegetation taking hold on the deglaciated landscape, classically considered processes (land-ocean exchange, biospheric uptake, etc.) reassumed dominant control on fluctuations of atmospheric chemistry. As a corollary of this hypothesis: $CO_2$ rise within the envelope of glacial-interglacial cyclicity would primarily have responded as a slave to orbitally controlled glacial retreat across kerogen impregnated landscapes as a result of global temperature increase. $CO_2$ rise may therefore lead or lag global temperature depending on the spatiotemporal patterns of glacial retreat that exposes glacially ground, kerogen-rich or even bituminous parent material.

However, in tandem with hypothesized kerogen oxidation, Earth system constraints such as the carbon isotope record dictate that other major processes must have acted. Various marine mechanisms have been proposed to explain $CO_2$ increases at glacial-interglacial transitions including the solubility pump hypothesis, iron fertilization hypothesis, and ocean circulation hypotheses (see hypotheses and reviews by Martin, 1990; Broecker and Peng, 1993; Kohfeld and Ridgwell, 2009; Rapp, 2019). As a cogwheel operating under manifold feedbacks in the greater Earth system (Sigman and Boyle, 2000), continuous glacial retreat, and the oxidation of finely ground kerogen, provide a hypothesis consistent with contributing to $CO_2$ increase in the wake of glacial episodes. This dilution of radiocarbon-dead $CO_2$ in the atmosphere may well have been complemented by other terrestrial sources such as the oxidation of subglacial paleosols and permafrost-bound organic carbon (Zeng, 2007; Simmons et al., 2016; Tesi et al., 2016; Crichton et al., 2016; Martens et al., 2020; Winterfeld et al., 2018; Lindgren et al., 2018; Köhler et al., 2014; Ciais et al., 2012) and by volcanic emissions triggered by deglacial unloading of the lithosphere (Roth and Joos, 2012). In addition to studies of weathering in glacial forefields and source-to-sink tracing of sedimentary kerogen, several lines of geochemical evidence including atmospheric carbon isotope composition ($^{13}C$ and $^{14}C$), which have thus far received contorted, partial explanations (Broecker and Clark, 2010; Schmitt et al., 2012; Broecker and McGee, 2013), glacial-interglacial changes in $^{14}C$ of DIC in seawater (Rafter et al., 2019, c.f., discussions therein), seawater osmium isotope changes, and long-term atmospheric $O_2$ content, conceptually go hand-in-hand with an opening of the exogenous kerogen cycle modulated by glacial activity. While geomagnetic variability and ocean ventilation together struggle to fully explain observed changes in atmospheric radiocarbon (Broecker and Barker, 2007; Cheng et al., 2018), the dilution of atmospheric $CO_2$ by accelerated oxidation of ancient terrestrial organic carbon at glacial terminations, in conjunction with other mechanisms

including atmosphere-ocean gas exchange (e.g., Sigman et al., 2010; Marcott et al., 2014; Menviel et al., 2018; Martin, 1990;

Sarntheim et al., 2013; Ai et al., 2020) appears as a simple and plausible explanation. In addition to kerogen oxidation at glacial terminations, the hypothesis presented carries other important implications, including the closure of the exogenous kerogen cycle during glacial periods (Fig. 2) potentially contributing to relatively high atmospheric $^{14}$C signatures, a problem highlighted by Dinauer et al. (2020), as a reduced $^{14}$C-free $CO_2$ flux would reduce the dilution of the atmosphere's cosmogenic $^{14}$C. Overall, increased reburial efficiency of kerogen can account for several tens of PgC over millennial timescales, entirely

bypassing actively circulating carbon pools on Earth's surface (Fig. 1). In contrast, increased oxidation efficiency of kerogen in the wake of glacial episodes that have built up stores of finely ground reactive substrate and exposed fresh weathering profiles can account for several hundreds of PgC over millennial timescales released into actively circulating pools. Owing to increased bedrock exhumation over the Ice Ages (Herman et al., 2013; Herman et al., 2015) the dynamism of the exogenous kerogen cycle may have been intensified, with greater fluxes of detrital kerogen reburied in ocean sediments during glacial

episodes and increased supplies of ground kerogen exposed to the elements in their wake.

While basic controls on kerogen reburial efficiency have emerged, its quantitative impact on atmospheric chemistry through geologic time remains conjectural. Mathematically analogous modeling results (Zeng, 2003; Simmons et al., 2016) suggest that kerogen oxidation could account for 30-60 ppm rise in atmospheric $CO_2$ over the course of the last deglaciation. Aligning with this, the back-of-the-envelope mass balance calculations presented in Table 1 suggest $CO_2$ release from kerogen and the

ocean each contributed around half to the total increase in atmospheric and terrestrial biospheric carbon pool sizes. However, there is a lack of studies on kerogen weathering to provide sufficient quantitative constraint for testing this hypothesis. To work towards such a test, investigating kinetics of kerogen oxidation along glacial chronosequences with contrasting lithologies would provide numerical input for Earth system models. Such information in conjunction with spatiotemporal changes in land ice extent (e.g., Dalton et al., 2020; Stroeven et al., 2016) integrating over areal changes of glacially exposed

lithological units (c.f., Copard et al., 2007) would constrain the quantitative impact of the exogenous kerogen cycle on atmospheric chemistry over glacial-interglacial cycles (e.g., see Fig. 2 in Kohfeld and Ridgwell, 2009). Additionally, the role of super carbon source terrains (e.g., surficial oil sands and coal) deserve special attention as biogeochemical weathering and associated $CO_2$ fluxes emanating from such areas conceivably contribute disproportionately, particularly upon deglaciation. Overall, the spatiotemporal deglaciation of contrasting source terrains across North America with the coeval progression of

atmospheric chemistry provide a strong incentive to explore the potential role of the exogenous kerogen cycle in glacial-interglacial patterns. From a greenhouse perspective, further study of pivotal episodes such as the Paleocene-Eocene Thermal Maximum under this lens may provide geological outlooks that are relevant today. In the context of our warming world, once critical thresholds are breached (Steffen et al., 2018), enhanced opening of the exogenous kerogen cycle may entrain the Earth system onto a new trajectory influencing the carbon cycle and climate for millennia to come.

## Acknowledgments

This work greatly benefitted from the input of several anonymous reviewers. The author is thankful for discussions with Timothy Eglinton, Dominik Letsch, Maarten Lupker, Jesper Suhrhoff, Valier Galy, Naohiko Ohkouchi, and Robert Hilton. This work was supported by funding from JAMSTEC.

## Competing interests

The author declares that he has no conflict of interest.

## Author contributions

TMB conceived of and wrote this contribution.

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

**Table 1: Carbon budget example illustrating a possible partitioning of carbon between pools mirroring the characteristic features of the stable carbon isotope trajectory of marine DIC and atmospheric $CO_2$ from the Last Glacial Maximum into the Holocene working with the idea of kerogen oxidation (see also Fig. 3). Numerical constraints from/modified after a) Ciais et al., 2012, b) Sigman and Boyle (2000), c) Schmitt et al., (2012), d) Peterson et al., (2014), e) Hedges and Oades (1997), f) Beaupré (2015), g) Meyers (1994), h) Still et al., (2003), i) Zeng (2003), and j) Simmons et al., (2016). *) The difference between Last Glacial Maximum and Holocene**
**DIC size was set at 520 PgC following Ciais et al., (2012), **) The Last Glacial Maximum global ocean value representing 0.5-5 km depth was used and 0.34‰ was added to represent the isotope shift into the Holocene (Peterson et al., 2014). Values rounded to the nearest 10 PgC and 0.01‰.**

| | Last Glacial Maximum | | Transition (ca. 12-15 ka BP) | | Pre-industrial Holocene | |
|---|---|---|---|---|---|---|
| | PgC | $\delta^{13}C$ [‰] | PgC | $\delta^{13}C$ [‰] | PgC | $\delta^{13}C$ [‰] |
| **Exogenous Carbon Pools** | | | | | | |
| Atmosphere $CO_2$ | 400[a] | -6.4[c] | 500 | -6.7[c] | 600[a,b] | -6.3[c] |
| Marine Carbon Pools | | | | | | |
| DIC | 38520[a*] | 0.11[d**] | 38260 | 0.28 | 38000[a,b] | 0.45[d*] |
| Dissolved OC | 700[e] | -22.5[f] | 700[e] | -22.5[f] | 700[e] | -22.5[f] |
| Terrestrial OC/ Soil | 1180 (total) | | 1640 (total) | | 2100[b] (total) | |
| C$_4$ | 560 | -14[g] | 500 | -14[g] | 420[h] | -14[g] |
| C$_3$ | 620 | -27[g] | 1140 | -27[g] | 1680[h] | -27[g] |
| | | | | | | |
| **Endogenous OC released** | | | | | | |
| Kerogen OC oxidized since Last Glacial Maximum | 0 | -25 | +300 | -25 | +600[i,j] | -25 |
| **Exogenous Carbon (Isotope) Mass Balance** | | | | | | |
| Without kerogen-derived OC | 40800 | -0.95 | 40800 | -0.95 | 40800 | -0.95 |
| With kerogen-derived OC | 40800 | -0.95 | 41100 | -1.12 | 41400 | -1.30 |

**Table 2: Fluxes of kerogen oxidation normalized to one year from onsite (soil and outcrops), catchment-wide studies (river), and laboratory-based incubation studies ordered approximately from small to large $CO_2$ release fluxes. For calculations, see online supplemental.**

| Area/ Material | Oxidation flux [MgC/km²/yr] | Comments | References |
|---|---|---|---|
| Andes | ≥0.3 | Kerogen oxidation in the Madeira floodplain based on radiocarbon isotope mass balancing of sedimentary organic carbon. Oxidation flux normalized to catchment area reported in Clark et al., (2017). | Bouchez et al., (2010) |
| | ~1.2 | Kerogen oxidation in the Madeira floodplain based on radiocarbon isotope mass balancing of sedimentary organic carbon refining the result by Bouchez et al. (2010). | Clark et al., (2017) |
| Mackenzie River Basin | $0.45^{+0.19}_{-0.11}$ | Rhenium-based estimate integrating over catchment and riverine in-situ weathering. | Horan et al., (2019) |
| Global average | $0.72^{+0.17}_{-0.41}$ | Calculated global average rate of kerogen oxidation on land. See Equation 1. | See text. |
| Soil, glacial till (Canada) | 4.2 | Holocene-averaged flux of onsite kerogen oxidation based on $CO_2$ evolution and carbon isotopes. | Keller and Bacon (1998) |
| Taiwan | 6.1-18.6 | Radiocarbon biogeochemistry reveals microbially-mediated weathering of kerogen onsite in soils. | Hemingway et al., (2018) |
| | 12±6 | Sedimentary organic carbon mass balancing with radiocarbon integrating over catchment and riverine in-situ weathering | Hilton et al., (2011) |
| | 7.4-13.0 | Rhenium-based estimate integrating over catchment and riverine in-situ weathering. | Hilton et al., (2014) |
| Shale (Posidonia, Germany) | 11-16 | Onsite kerogen oxidation based on Holocene scenario described in reference using mass loss of kerogen and estimated erosion rates. Weathering depth of 5 m and organic carbon to organic matter conversion factor of 1.5 assumed. | Littke et al., (1991) |
| Southern Alps, New Zealand | 9-50 | Rhenium-based estimate integrating over catchment and riverine in-situ weathering. Oxidation fluxes reported for 4 catchments with minimum and maximum reported here. | Horan et al., (2017) |
| Tar sandstone (Brazil) | ~54 | Laboratory-based experiments over 1 year duration under water saturated and anoxic conditions at 20°C. Results scaled with density taken as 2000 kg/m³ and 1 m weathering depth. | Rogieri Pelissari et al., (2021) |
| Shale (Jurassic marl, France) | 61-64 | Field-based observations spanning over minutes to up to 101 days using zeolite traps. | Soulet et al., (2018) |
| Bituminous coal | 320-530 | Laboratory-based subaquatic oxidation experiments at 24°C for air-saturated water over 179-442 days duration. Carbon release calculated assuming 30-50% of oxygen reacts to form $CO_2$, an average specific surface area of coal of 1 m²/g, average density of 1400 kg/m³, and 1 m weathering depth. | Chang and Berner (1998, 1999) |
| Shale (Brazil) | 80-1050 | Laboratory-based experiments over 75 days duration under water saturated, dark, and anoxic conditions at 25°C. Results scaled with density taken as 2200 kg/m³ and 1 m weathering depth. | Bertassoli Jr. et al., (2016) |
| Oil sands (Bitumen) | 6800-13000 | Laboratory-based experiments of 100 days duration for biodegradation of bitumen with different microbial cultures. Results scaled to oil sands with density taken as 2000 kg/m³ with 10% bitumen content, and 1 m weathering depth. | Ait-Langomazino et al., (1991) |

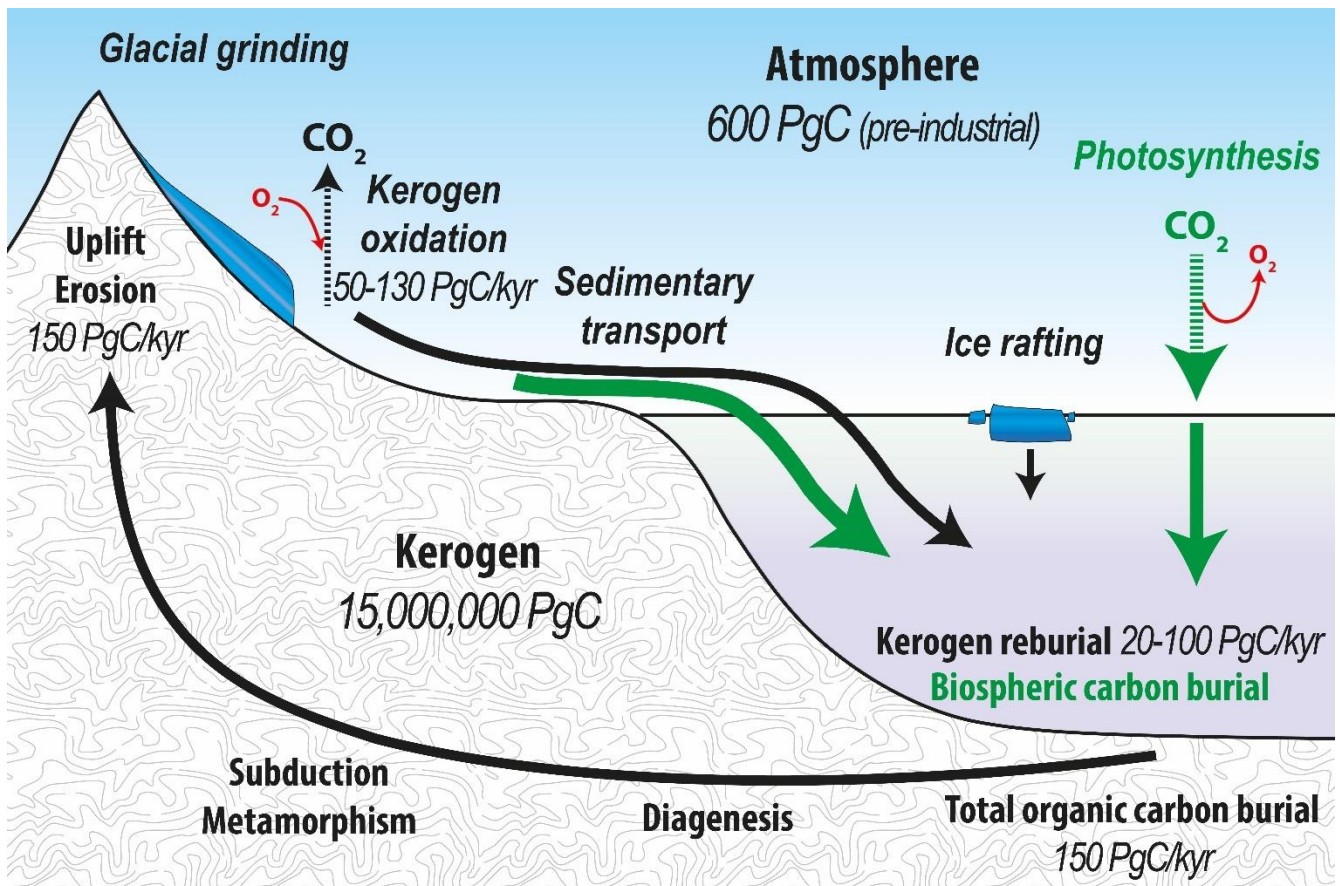

**Figure 1: Organic carbon cycle with the flow of kerogen (black solid lines) and the flow of biospheric carbon (green solid lines) showing the fixation of atmospheric CO₂ by both terrestrial and marine primary productivity. The combined flux of reworked kerogen and biospheric carbon into ocean sediments constitutes the total organic carbon burial into the endogenous kerogen pool (Galy et al., 2015; Hedges and Oades, 1997).**


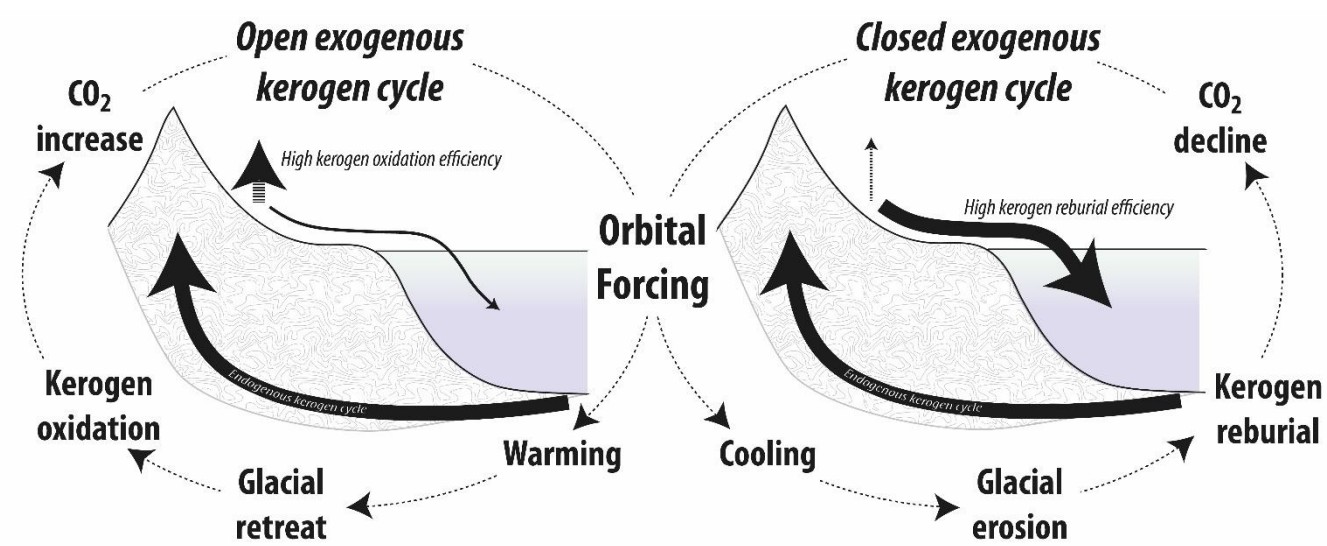

Figure 2: Conceptual hypothesis overview: Changes in kerogen reburial efficiency and its effect on the reentry of ancient carbon into surficial carbon pools as a function of overall climate state. During glacial times, kerogen reburial is promoted by the activity of glaciers and ice sheets with relatively little oxidation of this carbon during its transit across Earth's surface characteristic of a "closed" exogenous kerogen cycle. During glacial terminations and interglacials, the oxidation of kerogen is more efficient leading to the exhalation of this carbon to the atmosphere characteristic of an "open" exogenous kerogen cycle.

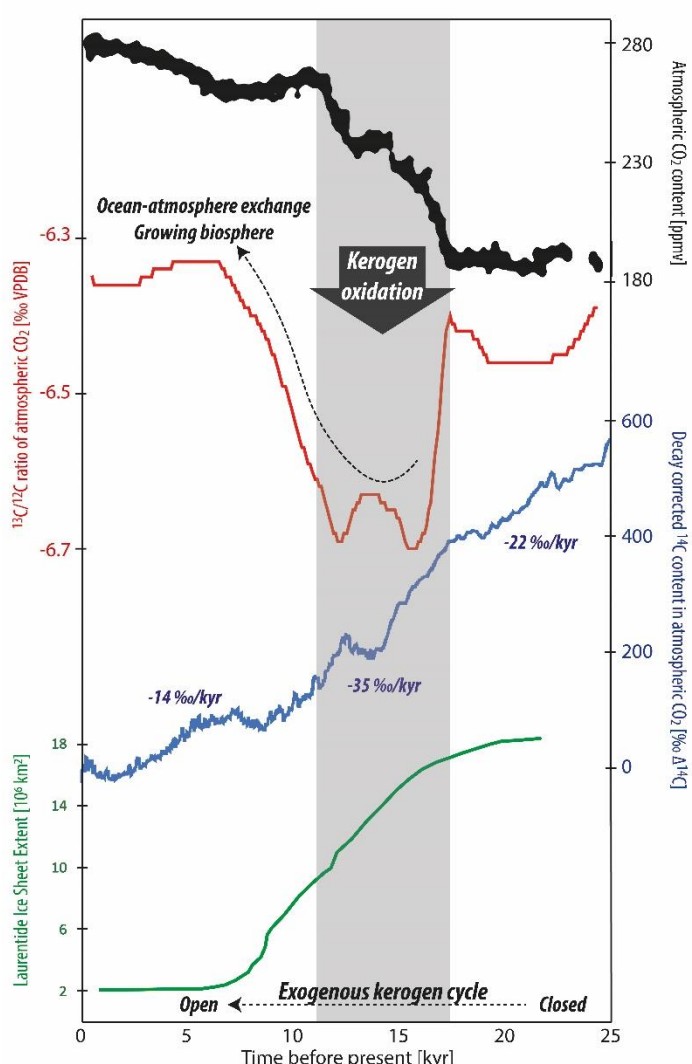

**Figure 3: Evolution of atmospheric CO₂ and its carbon isotope composition starting from the Last Glacial Maximum with redrawn data from Schmitt et al. (2012) and ¹⁴C data from Reimer et al. (2013). The area of the Laurentide Ice Sheet (including Greenland) is plotted after Dalton et al. (2020). The gray time envelope indicates the greatest rate of atmospheric CO₂ increase coinciding with a pronounced negative pulse in stable carbon isotope composition and accelerated decrease in radiocarbon concentration. In addition to the hypothesized role of kerogen oxidation, the superimposed effects of ocean-atmosphere CO₂ exchange and a growing terrestrial biosphere must have contributed in a major way to the evolution of these parameters.**


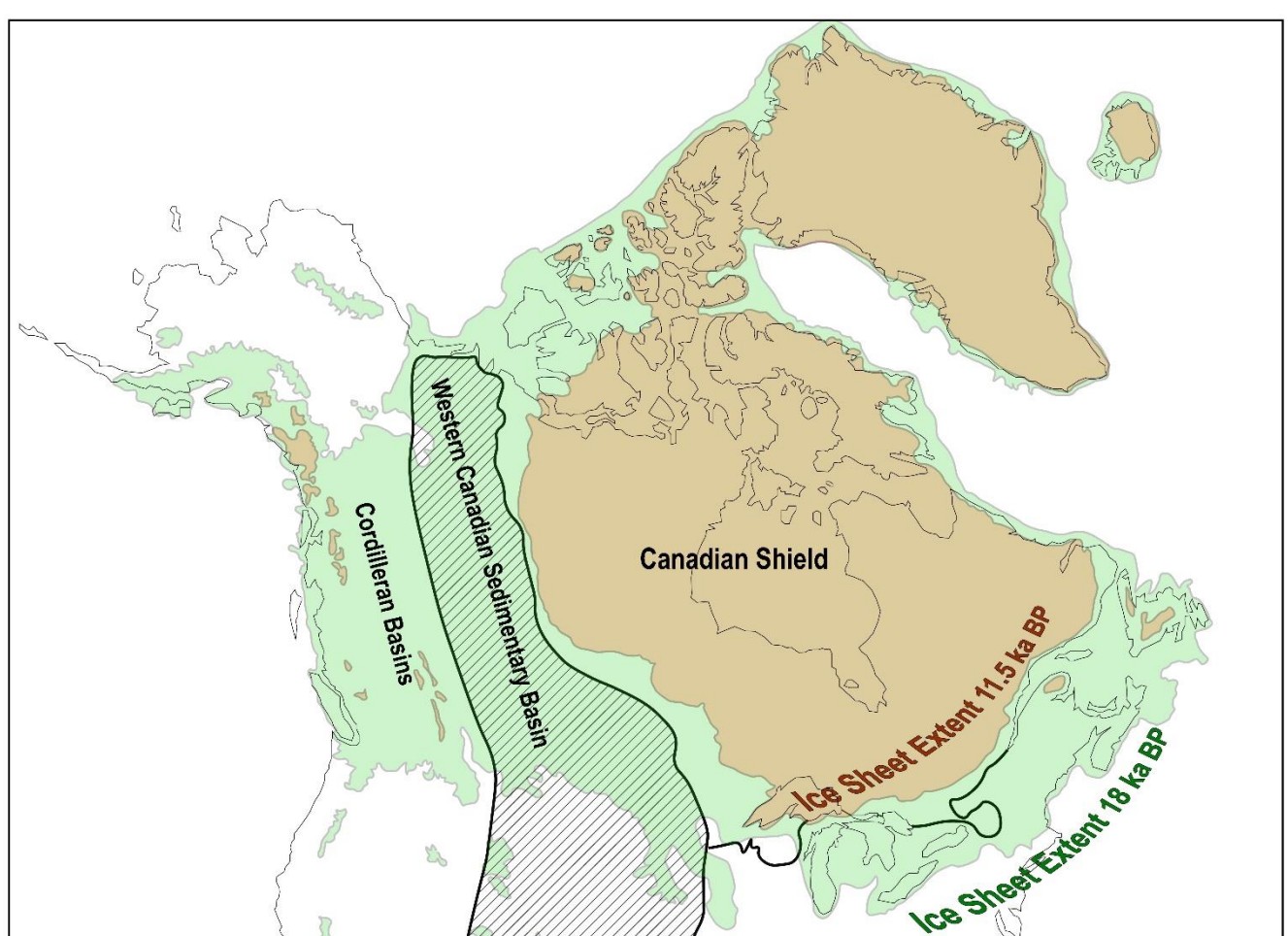

Figure 4: Laurentide Ice sheet extents bracketing the time span of most pronounced glacial-interglacial $CO_2$ increase (redrawn after Dalton et al., 2020). The Western Canadian Sedimentary Basin (hatched area, with extension of related units into the United States) and the Cordilleran Basins are to the west of the Canadian Shield (simplified after Miall and Blakey, 2019). The southern border of the Canadian shield is drawn with mainly Phanerozoic sedimentary units to the south (simplified after Reed et al., 2004). The surficial kerogen content in the Canadian Shield is generally low, while high kerogen contents are present along the axes of the Cordilleran and Western Canadian Sedimentary Basins (see Fig. 2 in Copard et al., 2007). ≤300 years after glacial retreat extended into the Canadian Shield, the rise in atmospheric $CO_2$ subsided (compare with Fig. 3). Here, kerogen oxidation in western Canada is hypothesized to have acted as a major source of $CO_2$ to the atmosphere in the direct wake of glacial retreat.