# Peer review of "Ideas and perspectives: Emerging contours of a dynamic exogenous kerogen cycle"

_Biogeosciences, 2021_

## Referee Comment (RC1)

**Review of:** "Ideas and perspectives: Emerging contours of a dynamic exogenous kerogen cycle"

**Author:** Thomas M. Blattmann

**Submitted to:** *Biogeosciences Discussions*

**Overall impression:**

This discussion manuscript presents a new hypothesis and a review of data that qualitatively support that hypothesis. The author argues that kerogen oxidation and burial efficiency are important mechanisms for modulating atmospheric $CO_2$ concentrations across glacial-interglacial periods (over $10^2$-$10^4$ year timescales). This is contrary to the common hypothesis that rock organic carbon oxidation and terrestrial organic carbon burial are important only over $10^5$-$10^6$ year timescales. Because we currently lack the data to sufficiently test this hypothesis, the author uses this manuscript to campaign for new studies to gather the datasets needed to improve our quantitative constraints on the feedback between kerogen oxidation and atmospheric $CO_2$ concentrations over glacial-interglacial cycles. This is an interesting hypothesis that should be given attention and the manuscript is well-written. However, there are several weak points that should be addressed.

I will also note that an earlier version of this manuscript was submitted in 2019, but was rejected for publication, largely due to lack of quantitative arguments and unconvincing discussion on the changes in weathering efficiency over time. This revised manuscript addresses most of the earlier reviewers' concerns, by adding some back-of-the-envelope calculations of potential atmospheric $CO_2$ changes due to kerogen oxidation, and a more thorough literature review and discussion. However, I think there are flaws in the quantitative argument, and I recommend substantial revisions before the manuscript can be accepted.

**Major points of concern:**

The author fails to bring kerogen oxidation into context with the other key processes modulating atmospheric $CO_2$ over glacial-interglacial timescales (e.g., silicate weathering, OC burial, changing biosphere). These processes are briefly mentioned, but should be acknowledged with a quantitative comparison (e.g., Hilton and West, 2020).

The author argues, rightfully so, that in the wake of glaciations, glacial retreat exposes kerogen-rich rocks and grinds them down, stimulating rock weathering and kerogen oxidation. This hypothesis is supported by decreasing atmospheric $^{14}C$ content from the LGM to present, which is consistent with input of radiocarbon-dead $CO_2$ to the atmosphere. The authors should elaborate on how erosion and weathering intensity changes across glacial-interglacial periods (e.g., Schachtman et al., 2019).

Deglaciation would also enhance carbonate weathering by the same physical breakdown mechanisms and subsequent meltwater dissolution, but the author argues that kerogen oxidation is a more important $CO_2$ source to the atmosphere than carbonate weathering during deglaciation, due to its faster weathering kinetics. The author support this argument with quantitative data (see Hilton and West, 2020).

Regarding the calculation made in Equation 1, the author overestimates the modern global average kerogen oxidation flux. They use a value of 150 PgC/kyr for kerogen exhumation, however, with a global fossil organic carbon stock of 1100 PgC (Copard et al., 2007) and a global average denudation rate of 5.4-6.5 cm/yr (Wittmann et al., 2020; Hedges and Oades,

1997), kerogen exhumation is at most 71.5 PgC/kyr. This agrees with the estimate of 40-100 PgC/kyr reported by Hilton and West (2020). Together, this would suggest that the modern kerogen reburial efficiency is ~60% and ~29 PgC/kyr is oxidized.

In this first equation, the author also uses kerogen oxidation rate estimates from Horan et al (2017), which were measured in the southern Alps of New Zealand. This setting is tectonically active, which enhances physical erosion and chemical weathering. As a result, the kerogen oxidation fluxes are overestimated and not likely representative of ice sheet retreat. In the case of the Laurentide ice sheet, the underlying lithosphere was passive, and physical erosion was only enhanced after glacial retreat induced isostatic uplift. While there are no other data for fully glaciated catchments, there are data for the Yukon and Mackenzie Rivers, which are likely more representative of large spatial scale kerogen oxidation fluxes in paraglacial conditions.

The author also fails to cite Hilton and West (2020), which is a key review paper that discusses the balance of $CO_2$ production and sequestration using data from river catchments around the globe. This manuscript could be improved by making a balanced carbon budgets for glacial periods and comparing them with those estimated for modern/interglacial conditions.

There are several physical and biological mechanisms that the author should address in this manuscript, to place kerogen oxidation into context with other mechanisms recognized to modulate atmospheric $CO_2$ over millennial-centennial timescales. See Schachtman et al. (2019) for physical and chemical erosion mechanisms over glacial-interglacial cycles, and perhaps Sigman and Boyle (2000) for quantitative insights to glacial-interglacial variability in biologic productivity.

The author does not consider the lag time between sediment production and export to the ocean. Presumably, upon glacial retreat, the pathway from glacier to ocean is short, and burial efficiency would overall be higher than today. However, sediment supply from glacial erosion is high, and much of the eroded material was deposited in moraines and glacial till, where it remains today. In the current manuscript, the author assumes that eroded material is largely delivered to the ocean and buried, but in reality this material can be stored for thousands of years during which it can be oxidized. If the author argues that atmospheric CO2 changes occurred within 300 years following glacial retreat, then kerogen oxidation must be very rapid. The author should consider transient sediment storage and potential lag times therein.

**Detailed comments:**

Lines 15-16:  The term "contributed majorly" doesn't really convey a clear message of how significant the increase in atmospheric $CO_2$ was as a result of deglaciation. It would be nice to give some estimate of the relative change in atmospheric $CO_2$ at the inflection point. If a more quantitative estimate is not feasible, then I suggest the author provide more context as to what other processes may have also contributed to the post-glacial increase in atmospheric $CO_2$.

Line 24: need reference for 15 million PgC kerogen

Line 29-31: Here, the author discusses the timescales over which kerogen oxidation and sedimentary organic carbon burial, mentioning that kerogen oxidation is important for atmospheric chemistry over million-year timescales, while sedimentary organic carbon burial is relevant over geological timescales. These timescales are apparently the same, so I think the second part of this sentence ("with kerogen oxidation considered important for

atmospheric chemistry over million-year timescales (e.g., Petsch, 2014; Bolton et al., 2006") should be moved to the end of the sentence on line 27. For example, "Upon oxidation of kerogen, $O_2$ is consumed and $CO_2$ is released to the atmosphere, *affecting atmospheric chemistry over million-year timescales.*"
Additionally, I'm unsure how the author can tie kerogen oxidation to atmospheric $CO_2$ changes over glacial-interglacial timescales when the relevant timescale for kerogen-atmosphere feedbacks is millions of years.

Lines 32-34: I would also re-word this sentence because kerogen decay can also be complete if organic-rich lithic fragments sit at earth's surface for a sufficient length of time such that the organic carbon is oxidized before being re-buried (e.g., Hemingway et al., 2018).

Lines 35-37: The author raises several questions to be addressed in this manuscript:
(i) what is the reburial efficiency of kerogen?
(ii) what is the weathering efficiency of kerogen?
(iii) what are their controlling factors?
(iv) what are the implications of them changing for atmospheric chemistry over geologic timescales?
In question (iv), the author should say "millennial/centennial timescales" rather than "geologic" because we generally know the implications over geologic timescales, as summarized by Petsch (2014). Their next sentence then presents the hypothesis that kerogen reburial and weathering efficiencies are important over centennial to millennial-scale atmospheric CO2 changes.

Line 37: Here, the author should highlight the overall knowledge gap, and emphasize how kerogen oxidation during glacial periods may be a key mechanism for changing atmospheric $CO_2$ concentrations across glacial-interglacial periods.

Line 42: clarify that export of organic matter and carbonate is from the surface ocean to the deep ocean or ocean floor

Lines 85 and 102: For the equations, the author should use variables in place of the numbers, then define the variables in the text. For example, rather than writing 149,000,000 $km^2$ in the denominator of equation 1, use the variable *A* for area. After describing the equations, then state what values or ranges of values were used to parameterize the equations, and finally the solution to the equation. This will make it easier for the reader to read and interpret.

Line 139: Is this supposed to read, "shales *and* oil sands"?

Lines 233-236: The author writes that the dilution of radiocarbon-dead $CO_2$ in the atmosphere could have been complemented by other terrestrial sources such as subglacial paleosol oxidation, permafrost-bound organic carbon oxidation, and by volcanic emissions due to unloading of the lithosphere.
Base on the cited literature therein, can the author make some estimates about the relative contributions of each of these processes to increasing atmospheric $CO_2$ in the wake of glaciation?

*References cited:*

Copard, Y., Amiotte-Suchet, P., & Di-Giovanni, C. (2007). Storage and release of fossil organic carbon related to weathering of sedimentary rocks. *Earth and Planetary Science Letters*, *258*(1–2), 345–357. https://doi.org/10.1016/j.epsl.2007.03.048

Hedges, J. I., & Oades, J. M. (1997). Comparative organic geochemistries of soils and marine sediments. *Organic Geochemistry*, *27*(7/8), 319–361.

Hemingway, J. D., Hilton, R. G., Hovius, N., Eglinton, T. I., Haghipour, N., Wacker, L., et al. (2018). Microbial oxidation of lithospheric organic carbon in rapidly eroding tropical mountain soils. *Science*, *360*(6385), 209–212. https://doi.org/10.1126/science.aao6463

Hilton, R. G., & West, A. J. (2020). Mountains, erosion and the carbon cycle. *Nature Reviews Earth & Environment*, *1*(June), 284–299. https://doi.org/10.1038/s43017-020-0058-6

Schachtman, N. S., Roering, J. J., Marshall, J. A., Gavin, D. G., & Granger, D. E. (2019). The interplay between physical and chemical erosion over glacial-interglacial cycles. *Geological Society of America | GEOLOGY*, *47*. https://doi.org/10.1130/G45940.1

Sigman, D. M., & Boyle, E. A. (2000). Glacial/interglacial variations in atmospheric carbon dioxide. *Nature*. https://doi.org/10.1038/35038000

Wittmann, H., Oelze, M., Gaillardet, J., Garzanti, E., & von Blanckenburg, F. (2020). A global rate of denudation from cosmogenic nuclides in the Earth's largest rivers. *Earth-Science Reviews*, *204*(February). https://doi.org/10.1016/j.earscirev.2020.103147

---

## Author Comment (AC1)

**Response to Reviewer Comment 1**

This discussion manuscript presents a new hypothesis and a review of data that qualitatively support that hypothesis. The author argues that kerogen oxidation and burial efficiency are important mechanisms for modulating atmospheric $CO_2$ concentrations across glacial-interglacial periods (over 102-104 year timescales). This is contrary to the common hypothesis that rock organic carbon oxidation and terrestrial organic carbon burial are important only over 105-106 year timescales. Because we currently lack the data to sufficiently test this hypothesis, the author uses this manuscript to campaign for new studies to gather the datasets needed to improve our quantitative constraints on the feedback between kerogen oxidation and atmospheric $CO_2$ concentrations over glacial-interglacial cycles. This is an interesting hypothesis that should be given attention and the manuscript is well-written. However, there are several weak points that should be addressed.

I will also note that an earlier version of this manuscript was submitted in 2019, but was rejected for publication, largely due to lack of quantitative arguments and unconvincing discussion on the changes in weathering efficiency over time. This revised manuscript addresses most of the earlier reviewers' concerns, by adding some back-of-the-envelope calculations of potential atmospheric $CO_2$ changes due to kerogen oxidation, and a more thorough literature review and discussion. However, I think there are flaws in the quantitative argument, and I recommend substantial revisions before the manuscript can be accepted.

Dear Reviewer,
Thank you for your constructive review. We indeed lack the background knowledge to test this hypothesis, and yes, the goal is to call to attention the potentially major role kerogen cycling on atmospheric chemistry, particularly with respect to the mystery of glacial-interglacial cycles. Responses to your concerns and feedback are given below. As a result of your input, the manuscript has greatly improved. Thank you very much.
Sincerely,
Thomas Blattmann

**Major points of concern:**

The author fails to bring kerogen oxidation into context with the other key processes modulating atmospheric $CO_2$ over glacial-interglacial timescales (e.g., silicate weathering, OC burial, changing biosphere). These processes are briefly mentioned, but should be acknowledged with a quantitative comparison (e.g., Hilton and West, 2020).

I agree. Context is now provided as a new paragraph starting off the section "3 Kerogen and glaciers – Dynamic modulators of the global carbon cycle?". The introduction is by design qualitative to avoid miring the reader's attention in the numbers and keep the reader focused on the core message of this work. However, to this end, the reader is referred to Hilton and West (2020) multiple times throughout the article. The important point emphasized is that mineral weathering (carbonate and silicate decay via carbonic and sulfuric acids) and biogeochemical processes (organic matter burial and kerogen oxidation) stand in close balance to one another over longer geologic timescales.

The author argues, rightfully so, that in the wake of glaciations, glacial retreat exposes kerogen-rich rocks and grinds them down, stimulating rock weathering and kerogen oxidation. This

hypothesis is supported by decreasing atmospheric 14C content from the LGM to present, which is consistent with input of radiocarbon-dead CO2 to the atmosphere. The authors should elaborate on how erosion and weathering intensity changes across glacial-interglacial periods (e.g., Schachtman et al., 2019).

Schachtman et al. (2019) is referenced as a comparison together with a list of other studies discussing changes in weathering across glacial-interglacial periods. Their study is in contrast to the studies cited for glaciated catchments. This is contained in this same new paragraph as mentioned above.

Deglaciation would also enhance carbonate weathering by the same physical breakdown mechanisms and subsequent meltwater dissolution, but the author argues that kerogen oxidation is a more important CO2 source to the atmosphere than carbonate weathering during deglaciation, due to its faster weathering kinetics. The author support this argument with quantitative data (see Hilton and West, 2020).

This is argued with the more primary references Horan et al., 2017 (and references therein) and Fischer et al., 2007.

Regarding the calculation made in Equation 1, the author overestimates the modern global average kerogen oxidation flux. They use a value of 150 PgC/kyr for kerogen exhumation, however, with a global fossil organic carbon stock of 1100 PgC (Copard et al., 2007) and a global average denudation rate of 5.4-6.5 cm/yr (Wittmann et al., 2020; Hedges and Oades, 1997), kerogen exhumation is at most 71.5 PgC/kyr. This agrees with the estimate of 40-100 PgC/kyr reported by Hilton and West (2020). Together, this would suggest that the modern kerogen reburial efficiency is ~60% and ~29 PgC/kyr is oxidized.

Firstly, Hilton and West (2020) report 40-100 PgC/kyr release from the oxidation of kerogen (please correct me if I am wrong). Also, Galy et al. (2015) suggest a detrital kerogen export of $43^{+61}_{-25} \frac{PgC}{km^2 yr}$, with uncertainty higher than the total kerogen exhumation estimate cited above, which would lead to a mass balancing problem. Additionally, kerogen exhumation and export is highly disproportionate and unevenly distributed (e.g., orogenic settings). Furthermore, the fossil organic carbon stock reported by Copard et al. (2007) integrates only over the top one meter of the earth surface. We know that kerogen oxidation starts taking place much below one meter (e.g., Petsch, 2014). Therefore, the kerogen exhumation rate suggested by the reviewer is an underestimation from the author's perspective. However, ultimately, reburial efficiency comes to lie in the same bracket (10-70%). The numbers today are poorly constrained and as the article argues: detrital kerogen export and oxidation fluxes varied through time, where our constraints are much poorer. Regardless of what numbers we choose the arguments in this "Ideas and Perspectives" article remain the same:

$F_{L-A}$ calculated with Eq. 1 will be a small number no matter what. This means that this has essentially no effect on the outcome of Eq. 2 and it makes no difference in terms of the overall picture of the magnitudes of the fluxes presented in Table 1.

In this first equation, the author also uses kerogen oxidation rate estimates from Horan et al (2017), which were measured in the southern Alps of New Zealand. This setting is tectonically active, which enhances physical erosion and chemical weathering. As a result, the kerogen oxidation fluxes are overestimated and not likely representative of ice sheet retreat. In the case of the Laurentide ice sheet, the underlying lithosphere was passive, and physical erosion was only

enhanced after glacial retreat induced isostatic uplift. While there are no other data for fully glaciated catchments, there are data for the Yukon and Mackenzie Rivers, which are likely more representative of large spatial scale kerogen oxidation fluxes in paraglacial conditions.

As the author argues, the author actually considers the oxidation rates by Horan et al. (2017) to represent an underestimate of the Laurentide Ice Sheet case for multiple reasons: one of which is the presence of "super carbon source terrains" in Western Canada with bituminous lithologies that show extremely high oxidation rates which are way beyond those of rock disseminated forms of kerogen (Table 1). This is a completely understudied aspect and definitely needs to be addressed in future research efforts.

The author also fails to cite Hilton and West (2020), which is a key review paper that discusses the balance of $CO_2$ production and sequestration using data from river catchments around the globe. This manuscript could be improved by making a balanced carbon budgets for glacial periods and comparing them with those estimated for modern/interglacial conditions.

After initially "failing", the author now successfully cites Hilton and West (2020) and quantitative-qualitative arguments are made while maintaining focus and flow for the reader.

There are several physical and biological mechanisms that the author should address in this manuscript, to place kerogen oxidation into context with other mechanisms recognized to modulate atmospheric $CO_2$ over millennial-centennial timescales. See Schachtman et al. (2019) for physical and chemical erosion mechanisms over glacial-interglacial cycles, and perhaps Sigman and Boyle (2000) for quantitative insights to glacial-interglacial variability in biologic productivity.

A plethora of physical and biological mechanisms are discussed throughout the article (which presents a completely original and highly interdisciplinary blend of literature!); an exhaustive review has already been provided by Hilton and West (2020). In contrast to a conventional review article, the readers reading this article are seeking new ideas and perspectives (hence the chosen article type) and this is what the author delivers: in a concise way with imaginative reasoning that will get a lot of people out of their comfort zones to go beyond textbook lines of thinking.

The author does not consider the lag time between sediment production and export to the ocean. Presumably, upon glacial retreat, the pathway from glacier to ocean is short, and burial efficiency would overall be higher than today. However, sediment supply from glacial erosion is high, and much of the eroded material was deposited in moraines and glacial till, where it remains today. In the current manuscript, the author assumes that eroded material is largely delivered to the ocean and buried, but in reality this material can be stored for thousands of years during which it can be oxidized. If the author argues that atmospheric $CO_2$ changes occurred within 300 years following glacial retreat, then kerogen oxidation must be very rapid. The author should consider transient sediment storage and potential lag times therein.

The author agrees that terrestrial redeposition of detrital kerogen in terrestrial environments are important for the exogenous kerogen cycle. This intermediate storage is constrained only loosely in a few regional settings (e.g., Blattmann et al., 2019b) and poorly constrained on a global scale (e.g., Meybeck, 1993). Transient storage should definitely be considered and this is expressed a couple of times with making recommendations for future research, "*Chronosequence studies of kerogen oxidation rates in deglaciated terrains are needed to provide constraints on time-integrated $CO_2$ release to the atmosphere.*" In response to the reviewer's comments, the following was added to emphasize the detrital kerogen reburial aspect of the problem: "*In*

*tandem with this, quantification is needed for (temporary) kerogen reburial in subaerial and subaquatic terrestrial systems (e.g., moraines, lakes) on global and regional scales (e.g., Meybeck, 1993; Vonk et al., 2016; Blattmann et al., 2019b; Fox et al., 2020)."*

The 300-year number is derived from the observed megascale spatiotemporal evolution of the Laurentide Ice Sheet as it retreats into the Canadian Shield with the timing of the inflection point in $CO_2$ increase. The author hypothesizes that kerogen oxidation happens continuously and parallel to glaciers retreating, the pedosphere transgressing, so everything is fluid an integrated perspective is needed to understand this number; this is the author's perspective hence the "Ideas and Perspectives" category of the article.

**Detailed comments:**

Lines 15-16: The term "contributed majorly" doesn't really convey a clear message of how significant the increase in atmospheric CO2 was as a result of deglaciation. It would be nice to give some estimate of the relative change in atmospheric CO2 at the inflection point. If a more quantitative estimate is not feasible, then I suggest the author provide more context as to what other processes may have also contributed to the post-glacial increase in atmospheric CO2.

I agree. However, as the sentence makes clear, this is hypothesis, and as the next sentence makes clear, quantitative constraints are needed. With the improvements made throughout the manuscript (e.g., adding in context with mineral weathering as elaborated previously above), the readers have more information to develop their own thoughts.

Line 24: need reference for 15 million PgC kerogen

Hedges and Oades (1997) and now the reference is moved to make it clear. Thank you.

Line 29-31: Here, the author discusses the timescales over which kerogen oxidation and sedimentary organic carbon burial, mentioning that kerogen oxidation is important for atmospheric chemistry over million-year timescales, while sedimentary organic carbon burial is relevant over geological timescales. These timescales are apparently the same, so I think the second part of this sentence ("with kerogen oxidation considered important for atmospheric chemistry over million-year timescales (e.g., Petsch, 2014; Bolton et al., 2006") should be moved to the end of the sentence on line 27. For example, "Upon oxidation of kerogen, O2 is consumed and CO2 is released to the atmosphere, *affecting atmospheric chemistry over million-year timescales.*" Additionally, I'm unsure how the author can tie kerogen oxidation to atmospheric CO2 changes over glacial-interglacial timescales when the relevant timescale for kerogen-atmosphere feedbacks is millions of years.

This is what the presented hypothesis is about. If this hypothesis motivates new research, future testing of this hypothesis will shed light on these ideas and perspectives. As the author argues throughout, there are several lines of strong, independent evidence that fit with this hypothesis. This work seeks to energize research interest in this direction.

Lines 32-34: I would also re-word this sentence because kerogen decay can also be complete if organic-rich lithic fragments sit at earth's surface for a sufficient length of time such that the organic carbon is oxidized before being re-buried (e.g., Hemingway et al., 2018).

In this section of the article, the author would like to keep the context on a global perspective. More local considerations are delved into later in the manuscript. As the sentence starts with a "however", it implies that previous studies often considered this to be the case. More often than

not, kerogen oxidation is incomplete (e.g., Hemingway et al., 2018; Leythaeuser 1973, and many more).

Lines 35-37: The author raises several questions to be addressed in this manuscript: (i) what is the reburial efficiency of kerogen? (ii) what is the weathering efficiency of kerogen? (iii) what are their controlling factors? (iv) what are the implications of them changing for atmospheric chemistry over geologic timescales? In question (iv), the author should say "millennial/centennial timescales" rather than "geologic" because we generally know the implications over geologic timescales, as summarized by Petsch (2014). Their next sentence then presents the hypothesis that kerogen reburial and weathering efficiencies are important over centennial to millennial-scale atmospheric CO2 changes.

Thank you for this constructively critical comment. The author has reformulated question (iv) in different direction: (i) what is the reburial efficiency of kerogen, (ii) what is the weathering efficiency of kerogen, (iii) what are their controlling factors, and (iv) how do reburial and weathering efficiency vary over geologic time and space?

The review by Petsch (2014) was given insufficient credit in this contribution and is now referenced in section 5 "Tackling geologic deep time" to highlight these contributions. Overall, the author is of the opinion that kerogen cycling (whether on geologically "short" or "long" timescales) is understudied, with very little primary data extending back in geologic time.

Line 37: Here, the author should highlight the overall knowledge gap, and emphasize how kerogen oxidation during glacial periods may be a key mechanism for changing atmospheric CO2 concentrations across glacial-interglacial periods.

The knowledge gap and apparent contradictions in the existing body of literature gets addressed in the next section. However, I have added a transitional sentence to make the transition of ideas smoother. Thank you.

Line 42: clarify that export of organic matter and carbonate is from the surface ocean to the deep ocean or ocean floor

Thank you. This has been fixed.

Lines 85 and 102: For the equations, the author should use variables in place of the numbers, then define the variables in the text. For example, rather than writing 149,000,000 km2 in the denominator of equation 1, use the variable *A* for area. After describing the equations, then state what values or ranges of values were used to parameterize the equations, and finally the solution to the equation. This will make it easier for the reader to read and interpret.

Thank you for pointing this out. I have formalized the equations with variables defined in the text.

Line 139: Is this supposed to read, "shales *and* oil sands"?

Yes. Thank you. Corrected.

Lines 233-236: The author writes that the dilution of radiocarbon-dead CO2 in the atmosphere could have been complemented by other terrestrial sources such as subglacial paleosol oxidation, permafrost-bound organic carbon oxidation, and by volcanic emissions due to unloading of the lithosphere. Base on the cited literature therein, can the author make some estimates about the relative contributions of each of these processes to increasing atmospheric CO2 in the wake of glaciation?

At the moment, our quantitative constraints are too rudimentary. Even the quantitative constraints for today's carbon cycle are still "emerging" as explicitly mentioned by Hilton and West (2020). However, based on the cited modeling studies, the following is stated in the

manuscript: "With this, a plausible scenario for releasing kerogen-derived $CO_2$ to the atmosphere that could account for a 30-60 ppm rise during the glacial-interglacial transition encompassing an area equal to or less than the terrestrial extent of the Laurentide Ice Sheet is identified." In the author's opinion, this is the best we can say at the moment for how much kerogen oxidation may have impacted atmospheric $CO_2$ rise during deglaciation. Due to the limited uniqueness of the geochemical parameters (e.g., $^{13}C$ and $^{14}C$ for permafrost and kerogen) simple geochemical models will not suffice in deconvolving the source mechanisms (auxiliary lines of clues however do point towards kerogen, as elaborated in the manuscript). Therefore, the author suggests basic research directions as elaborated in the text on how to proceed. All in all, the megascale spatiotemporal trends in the deglaciation of North America across the geologic boundary between the Canadian Shield and the adjacent sedimentary basins suggest a connection – a compelling piece of evidence that has seemingly gone overlooked!

---

## Author Comment (AC2)

**Response to Reviewer Comment 2**

I found this a novel, interesting, and generally well-written paper that argues that weathering of kerogen-containing lithologies exposed at the surface after continental deglaciation may prove to be a significant source of carbon dioxide to the atmosphere, and one which is of particular significance in terms of climate forcing. Whilst the argument is supported more by calculations and logical arguments than it is by direct measurements and observations, I still found it fairly compelling – to the point that I am convinced that the idea is worth pursuing via in situ measurements and carefully designed and executed experiments. It is certainly worth publishing if only to give exposure to the idea and to stimulate discussion and field monitoring of natural carbon emissions from kerogen sources as well as to provoke detailed modelling of likely CO2 fluxes from kerogen sources on geologically and climatically relevant timescales (and detailed mapping (in time and space) of likely source regions for kerogen-derived greenhouse gas emissions). Some articulation of likely important source regions for such emissions would be a valuable contribution to the paper and the broader scientific discussion that it is likely to stimulate. It is certainly a paper that gave me a kick and made me challenge my prior assumptions and thinking about climate/greenhouse gas emission linkages. On that basis I think it is worthy of publication, although, at the detailed level, I think the text needs a thorough edit. Below I have provided a set of suggestions that I hope might help with this.

Dear Reviewer,

Thank you for your thorough and constructive review. It is surprising how poorly constrained $CO_2$ fluxes emanating from kerogen are. Monitoring of such would provide valuable baseline data for understanding our Earth system. And I agree, articulating the possible source region (western Canada) is useful for pinpointing our discussion and focusing future research efforts.

Finally, I am so happy that this article gave you a "kick"! This novel idea presented with a blend of interdisciplinary literature in this "Ideas and Perspectives" format provides a platform to challenge and progress our thinking!

Sincerely,

Thomas Blattmann

**Line by Line Review (i.e. suggested changes to the text that I think would improve it's readability and clarity):**

9: suggests that this largest pool

Corrected. Thank you.

10: interglacial cycles and beyond

Corrected. Thank you.

15: in western Canada contributed in a major way

Corrected. Thank you.

25: subjected 150 PgC/kyr.....

This is present tense as it is an ongoing process. This sentence has been simplified following the next comment.

26: of this geologically ancient carbon and other closely connected surficial carbon pools into the atmosphere (Hedges and Oakes, 1997)

This sentence has been simplified to make it more readable. Thank you.

30: compensatory roles

Thank you for your correction.

32: as physical erosion is followed by riverine transport

This is part of a subclause, so I think the grammar is correct.

37: This contribution hypothesizes…..atmospheric CO2 increases during glacial terminations

Thank you for your suggestion and correction. Implemented.

42: and (4) the export of organic matter and carbonate from the surface waters of the oceans - Question – export to where?

This has now been specified. Thank you:

… from the surface to deep waters and sediments of the oceans …

43: During deglaciation

Thank you for the suggestion. Implemented.

44-45: an increasingly voluminous terrestrial biosphere (but is it mass or volume that matters here?)………is inferred to have controlled an increase in the stable carbon isotope ratio of dissolved organic carbon in ocean waters.

Here, I did mean dissolved inorganic carbon. This should be correct as is.

In order to avoid misunderstanding, the reference to volume has been removed and improved to: "…an increasingly large terrestrial biospheric carbon pool …". Thank you.

46: carbon pools changing in size at the same time as stable carbon isotope fractionation occurs, as carbon is exchanged between pools such as the terrestrial biosphere and pedosphere (see also Zeng, 2003,2007)

Thank you. Your suggestion has been fully implemented.

47: In addition, during times of most rapid CO2 increase during transitions from glacial to interglacial periods, negative stable carbon isotope shifts in atmospheric CO2 occurred (Fig.3; Smith et al., 1999; Schmitt et al., 2012).

Thank you for these corrections. Helps a lot to receive these.

49: This is a strong indicator that respired organic carbon was acting as a direct source to the atmosphere (Bauska et al., 2016).

Thank you. This was implemented with a small modification: This is a strong indicator that respired organic carbon acted as a direct source to the atmosphere (Bauska et al., 2016).

51: that was depleted in or devoid of radiocarbon………thereby limiting the potential contributions from a modern biospheric organic carbon source. BUT does it actually limit the contributions, or just their detectability?

In my understanding, it does limit the contribution size, because biospheric carbon and kerogen-derived carbon represent extreme end members in natural abundance $^{14}C$. Detectability is another matter, but the trends in atmospheric chemistry (CO2 ppm, d13C, D14C) are quite large.

53-54: deep ocean was the predominant source for carbon transferred to the atmosphere during glacial terminations

Thank you for these corrections. Implemented.

55-56: please explain what you mean by "requires a complex overlay of processes to reconcile"

To maintain flow in the text, I have simplified the statement to: However, this hypothesis appears inconsistent with the negative fluctuation observed in the 13C fingerprint of atmospheric CO2 (see discussion in Broecker and McGee, 2013).

58-59: suggest that the release, via kerogen oxidation, of CO2 to the atmosphere during deglaciation contradicts or complements the commonly held notions of a strictly increasing terrestrial organic carbon pool and major changes in CO2 exchange between the ocean and the atmosphere.

This sentence has been completely rearranged.

58-60: needs some supporting references

References have been added. Thank you.

62: accumulated from….supports the idea that…..was more extensive

Thank you very much. Corrected.

63: cold interludes in Earth history during which glacial erosion and ice rafting dominated (BUT – what did they dominate?)

I replaced this with a better phrase: "was widespread". Thank you.

64: reburial in high latitude glaciated regions…

Thank your for this improvement.

66: kerogen cycle by keeping…..

Thank you. Corrected.

69: frost shattering, together with the retreat of glaciers, exposes……thereby accelerating oxidation and the release of kerogen-derived CO2…….declines into an interglacial period.

Thank you. All of these points have been corrected.

73. Analogously, glaciers have also been invoked as agents for accelerating chemical weathering of carbonate and silicate minerals by increasing sediment yield and creating a reactive substrate with high surface area. Carbonate weathering can be a source of CO2 to the atmosphere when sulphuric acid is the solvent involved. (I assume this is a by product of sulphide mineral (pyrite) oxidation? Please clarify this)

Thank you for these improvements, and yes, this is the byproduct of sulfide mineral oxidation.

77. direct conversion to CO2 leads to considerable….

Thank you for this improvement.

78-79: This is a process by which CO2 can be injected directly into the atmosphere and impact glacial-interglacial cycles (Figure 2)

Thank you for this improvement.

90: faster than those of the average Earth surface

Thank you. Implemented.

95: also proposes the oxidation of overridden soil organic carbon during and after glaciation and calculates a 600 PgC release….

Thank you for your careful reading. I have implemented a modified version of your suggestion. In my opinion, XX et al. refers to multiple authors, so is grammatically equivalent to "they": Similarly, modeling by Simmons et al. (2016) also propose the oxidation of overridden soil organic carbon subsequent to deglaciation…

115: fluxes an order of magnitude greater than the global average

Thank you. Implemented.

120-127: Are the kerogen oxidation and oceanic release mechanisms for CO2 increase mutually exclusive? You make it sound as though they are, but I'm not clear why that would be the case.

Thank you for pointing this out. I have fixed this area by removing a reference that was very poorly chosen by me which alluded to oceanic release in a confusing way. The text is now

straightforward and of course they (kerogen oxidation and oceanic $CO_2$ release) operate independently of one another.

115: oxidation fluxes an order of magnitude greater than the global average can be sustained for millennia after deglaciation.

Thank you. I have implemented this.

134: extending across much..

Thank you. Corrected.

137: within the Province of Alberta

Thank you. I have corrected this.

139: Cretaceous soils and the oil sands…….the latter enhanced by aerial exposure across palaeosurfaces

Thank you. I have corrected this.

140: over tens of thousands

Corrected. Thank you.

145: Laboratory incubations designed to simulate $CO_2$ respiration from bituminous materials reveal fluxes that are markedly higher than those associated with oxidation of rock disseminated forms of kerogen (Table 1)

Thank you for this improvement.

147-148: at rates 1-2 orders of magnitude higher than those reported for rock disseminated kerogen, and 3 orders of magnitude greater than the average for Earth's surface.

Thank you for these corrections. Implemented.

152: when temperatures of subaerially exposed outcrops of oil sands reach 60°C

Thank you for this improvement.

153: experiments on bitumen

Thank you. Corrected.

155-156: that investigated the oxidative decay of hydrocarbon fractions also suggest similarly high fluxes when scaled to natural systems, even though these studies were conducted over periods of only a few weeks

Thank you for this improvement.

158: fluxes reported by Chang and Berner (1998,1999)…an underestimate

Thank you for these corrections.

160: $CO_2$ can be released under anaerobic conditions

Thank you this has been corrected.

162-163: what is meant by a super-carbon source terrain? Maybe useful to identify some specific examples

I have added a hypothetical definition to the concept of: "super carbon source terrains": *areas laced or covered with coal, bituminous materials, etc.*

163: during glacial-interglacial transitions. This statement makes me wonder whether you have given any thought to what happens in interglacial-glacial transitions. Are you just assuming that overriding by ice shuts off exchanges between substrate and atmosphere – but would that necessarily preclude gas transfer through permeable substrates along the hydraulic potential gradient from thick ice in the interior to thin ice at the margins where gas could escape to the atmosphere?

*I have improved the wording in this part. Previously, it read like there was a conflict, where there was none. Now the paragraph ends with: Overall, it is conceivable that super carbon source terrains (hypothetically, areas laced or covered with coal, bituminous materials, etc.) across western Canada could supply an overproportionate quantity of radiocarbon dead CO2 to the atmosphere during glacial-interglacial transitions.*

164-5: Sheet had retreated…..and was exposing

Thank you for this correction.

167: <= 300 years after….Sheet advanced onto the Canadian Shield, suggesting reduced decay of…

Thank you. I have adopted a change modified after your suggestion.

170: Fennoscandian Ice Sheet

Thank you for this correction.

173: is chemically recalcitrant

Thank you for this improvement.

175: was the most extensive element of the cryosphere that waxed and waned across the continents…….and, in conjunction with its lithological underpinning…2007),

Thank you for these improvements.

178: estimates of CO2 fluxes…….and there is considerable uncertainty in our current state of knowledge

Thank you for these corrections.

179-180: weathering studies that provide estimates of CO2 fluxes from bedrock-derived kerogen under relevant environmental conditions and over appropriate timescales are lacking

Thank you for these improvements. Implemented.

182: high resolution reconstructions of changes in land ice extent and the lithologies of bedrock and glacial till being exposed by glacial retreat can, in theory, quantitatively disentangle the contribution of kerogen-derived CO2 to the atmosphere during glacialinterglacial transitions
A question here – can isotopic fingerprinting methods distinguish between the kerogenderived CO2and CO2 from other potential sources?

Thank you for this correction. Implemented.

Other organic sources can only distinguished if they contain radiocarbon. In the case of old permafrost, this may not be the case. However, detrital kerogen redeposition, osmium isotopes, rhenium, iodine isotopes, etc. offer complementary tools to disentangle the carbon isotope record.

189-90: Also important are the bedrock lithology and regolith composition

Thank you for this correction.

192: increasingly suggests that…

Thank you for this correction.

194: increased the flux……..Over Earth's history, on 109 year timescales the reburial efficiency of kerogen presumably varied….

Thank you. These corrections have been implemented.

198-199: O2 on 106 year timescales

Thank you. Corrected.

200: to understand changes in atmospheric chemistry through geologic time…

Corrected. Thank you.

201: the changing efficiency of the reburial of kerogen needs to be evaluated

Thank you for your improvement.

204: geospatial variability in what ?

This has been improved to: "uneven spatial dispersal". This captures the meaning much better in referring to the sedimentation behavior.

205: for quantifying, and establishing the importance of the reburial of kerogen in recent times, it's utility diminishes quickly for strata that pre-date the Last Glacial Maximum owing to it's radioactive decay.

Thank you for these changes. I have implemented modified changes.

210: isotopic shifts at the beginning of interglacials that are attributable to kerogen oxidation…………

Thank you. I have made modified improvements.

211: consistent with the hypothesis presented here

Thank you for this improvement.

216: the hypothesis presented proposes

Thank you. Implemented.

218: the rate of decrease of 14C $CO_2$ subsided……..mirrored by changes, during deglaciation, in the lithologies of the Canadian Shield that were exposed at the surface, which contain relatively minor amounts of reactive kerogen.

Thank you for these improvements. I have adopted modified changes.

220-…The coincidence in time of global trends in atmospheric chemistry with spatiotemporal patterns in the distribution of freshly deglaciated terrain……suggests that a burst (or bursts) of respired $CO_2$ contributed to the characteristic deglacial increase in atmospheric $CO_2$.

Thank you. I have implemented these improvements.

224: soil and vegetation taking hold on the deglaciated landscape

Thank you.

228: patterns of glacial retreat that expose glacially ground, kerogen-rich or even bituminous parent material.

Thank you.

230: have been proposed to explain $CO_2$ increases

Thank you.

232: retreat, and the oxidation of finely ground kerogen, provide……

Thank you.

234: such as the oxidation of subglacial paleosols and permafrost-bound organic carbon….and by volcanic emissions triggered by deglacial unloading of the lithosphere

Thank you.

243-4: accelerated oxidation of ancient terrestrial organic carbon at glacial terminations…

Thank you.

246-7: the hypothesis presented…

Thank you.

250: timescales, entirely….

Thank you.

252: exposed fresh weathering profiles….

Thank you.

255-6: and increased supplies of ground kerogen

Thank you.

268: provide a strong incentive

Thank you.

269: kerogen cycle in glacial-interglacial climate patterns

Thank you. I would however like to avoid the word climate right here in direct connection with kerogen cycle. As Hilton and West (2020) state: "$CO_2$ sources might also be sensitive to climate indirectly through facilitation of oxidative weathering by glacial processes." I agree with this and this is also expressed in Figure 2.

270-271: may provide an outlook for geological processes that is relevant today

Thank you.

271: (Steffen et al. 2018) is missing from the reference list

Thank you for checking.

Figure 1 caption: showing the fixation of atmospheric CO2 by both terrestrial and marine primary productivity……..constitutes the total organic carbon burial into the endogenous kerogen pool.

Thank you for all these excellent improvements and careful corrections. The manuscript has greatly improved as a result of your hard work.

---

## Author Comment (AC3)

**Response to Reviewer Comment 3**

This work hypothesizes that de-glaciation and weathering of kerogen-rich lithologies in western Canada made a major contribution to CO2 rise at glacial terminations by compiling and reinterpreting empirical evidence. I have several comments regarding the interpretation and methods of the manuscript, which I hope to help to improve the manuscript. I'd recommend revisions before acceptance for publication.

**Dear Reviewer,**

Thank you very much for your critical and constructive review. I disagree with a few of the raised points. In the interest of idea flow and concise messaging, I have opted to keep the manuscript short - and this is exactly what gives readers a kick, just like with reviewer 2. My detailed responses are given below.

For decades, modelers have twisted ocean circulation and ensuing ocean-atmosphere exchanges into all sorts of pretzels to explain glacial-interglacial cyclicity and its fascinating patterns of atmospheric chemistry. Various hypotheses invoking terrestrial mechanisms have been presented in the literature with little breakthrough in our greater scientific progress. The faithful rise in atmospheric CO2 at glacial terminations has in many ways been a mystery that has evaded satisfactory explanation as highlighted multiple times by W. Broecker and others. This *"Ideas and perspectives"* article brings a radically new mechanism to the table supported by simple, first principal arguments. Clearly, quantitative constraints are rudimentary and are in need of refinement and many questions are far from being answered. To work towards this, the author hopes that this article will energize a new community effort to research the exogeneous kerogen cycle to inquire how dynamic it truly is.

Overall, clearly much more research needs to be undertaken to answer the many questions that rise from this "Ideas and Perspectives" article. Through the review process and thanks to your comments, this work has improved greatly. Thank you very much.

Sincerely,

**Thomas Blattmann**

**Major comments:**

Comment 1: 'kerogen' should be clearly defined: how does this term compare to the other terms used in the relevant studies, for example, rock-derived organic carbon, and petrogenic carbon. Do those concepts overlap/differ or are they the same? Would organic carbon in metamorphic rocks also be termed 'kerogen'?

Kerogen is indeed a term used quite differently by different authors. Durand (1980) probably provides the most "mainstream" definition with kerogen being a term reserved for insoluble rock-disseminated forms of organic matter. Geochemists who started using the term in the 2000s

along with a slew of other terms: "In literature, petrogenic OC is also referred to as fossil carbon, geogenic carbon, rock(-derived) carbon, ancient carbon, relic carbon, detrital carbon, and kerogen" (Blattmann et al., 2018). In biogeosciences (the field of the journal this contribution is submitted to), they are mostly used interchangeably. In this contribution, the author opts for kerogen as it is a single word that requires no abbreviation and is instantly recognizable across a wide spectrum of Earth science disciplines. A definition has been added as a footnote in the beginning to clarify the author's usage of the term: *In this work, kerogen is used as an umbrella term for all rock-derived forms of reduced carbon including soluble, insoluble, rock disseminated, rock forming, solid and liquid forms, as well as fossil palynomorphs, biogenic and abiogenic graphite.*

Durand, B.: Sedimentary organic matter and kerogen. Definition and quantitative importance of kerogen, in: Kerogen: Insoluble organic matter from sedimentary rocks, edited by: Durand, B., Éditions Technip, 13-34, 1980.

Comment 2: it needs to acknowledge existing understandings/studies about carbon fluxes during glacial-interglacial time periods. What did previous work find/conclude about glacial terminations and CO2 rise? How does the kerogen weathering hypothesis in this manuscript differ from the previous studies? What were the magnitudes of other carbon fluxes during deglaciation (e.g. the carbon exchange between the atmosphere and ocean reservoirs) and how do those compare to the kerogen weathering flux?

Fluxes of carbon between atmosphere-terrestrial biosphere and atmosphere-ocean are several orders of magnitude greater than any pathways of mineral weathering or kerogen oxidation fluxes. However, these are also compensated with nearly equal reverse fluxes across space and annual timescales (e.g.,  $CO_2$  uptake and  $CO_2$  release from different parts of the ocean, e.g., Menviel et al., 2018; Takahashi et al., 1997). References to studies championing different hypotheses of CO2 release pathways are concentrated in lines 260-286. The magnitudes of carbon fluxes during deglaciation have been subject of great many studies and it is clear that all of these studies leave basic observed parameters unexplained (see e.g., Broecker & Barker, 2007). This contribution introduces a new, simple, and plausible mechanism into this longstanding discussion, which is able to reconcile many of these parameters and also have a clear physical basis. I envision that this work will serve as foundation for new discussion and iterating improvement of our understanding of the global carbon cycle so that we can work towards a quantitative understanding, beyond the isolated processes, deconvolving the individual components acting in the integrated Earth system (e.g., Kohfeld and Ridgwell, 2009). With this, figure 2 in Kohfeld and Ridgwell (now referenced in the article) is the ideal goal of what would be desirable to achieve.

Takahashi, T., Feely, R. A., Weiss, R. F., Wanninkhof, R. H., Chipman, D. W., Sutherland, S. C., and Takahashi, T. T.: Global air-sea flux of CO2: An estimate based on measurements of sea–air pCO2 difference, Proceedings of the National Academy of Sciences, 94, 8292, 10.1073/pnas.94.16.8292, 1997.

Comment 3: there's no detailed lithology/kerogen amount/kerogen weathering kinetics/kerogen 13C signal/topography information (e.g. maps or data compilation) about western Canada – all

those variables are important and relevant to the total oxidation flux of kerogen, and need to be discussed.

Table 1 contains all the kerogen weathering kinetics available in the literature. This is the product of 30 years of research and it is compiled here for the first time representing the most thorough review of kerogen oxidation fluxes known to the author. Stock estimates of kerogen on Earth's surface are reported by Copard et al. (2007) and provide clear support for the hypothesis. However, the data by Copard et al. are too coarse spatially and need refinement to allow for quantitative application in models. A general reference is now added regarding the  $\delta^{13}$ C isotope composition of organic matter in rocks (Lewan, 1986), and it is also clear that glacial till composition is also important for which there is a lack of data. Compiling these maps (with  $\delta^{13}$ C) is beyond the scope of the current work. For now, it is important to know that kerogen is isotopically light as has been documented extensively for major sedimentary units through geologic time (Lewan, 1986) and for different metamorphic grades (e.g., Hoefs and Frey, 1976). Calculating total and time resolved oxidation flux of kerogen requires more parameters than we have available. Concerted research efforts are needed to deconvolve glacial-interglacial effects on kerogen oxidation (systematically extending the 30 years of data presented in Table 1). This would be desirable for any mechanism (e.g., atmosphere-ocean exchange), but model estimates are the best we have for these (as elaborated in previous answer). However, the paper outlines a research pathway so that this can be achieved in the future as enunciated in the abstract and conclusions. The possibility of kerogen oxidation as a key driver of CO2 increase at glacialinterglacial transitions is enunciated for the first time in this paper. Plausible arguments support this, and this paper seeks to energize research in this new direction.

**Hoefs, J., and Frey, M.: The isotopic composition of carbonaceous matter in a metamorphic profile from the Swiss Alps, Geochimica et Cosmochimica Acta, 40, 945-951, 1976.**

Comment 4: over glacial-interglacial timescales, would weathering of aged soil organic carbon (with residence time of thousands to tens of thousands of years) play an important role for CO2? Was aged soil organic carbon considered a part of kerogen in this work? How did the aged soil carbon flux/pool compare to those of kerogen organic carbon?

The soil hypothesis is championed by other works (e.g., Zeng, 2003). This work champions rockderived carbon. I personally disagree with the soil hypothesis as a large subglacial storage of soil organic matter is required (c.f., Lindgren et al., 2018). While some soil is certainly overridden by glaciers, kerogen supply from bedrock is virtually limitless as long as it is exhumed, making supply easy to explain. The Earth system is underdetermined, so deconvolution is riddled with unsatisfactory uncertainties; even a single isolated Earth system component is challenging to model. Soil and kerogen oxidation signatures on atmospheric chemistry are equivalent in terms of atmospheric chemistry for  $\delta^{13}$ C and direct conversion to CO2, therefore, modeling work by Zeng (2003) and Simmons et al.'s (2016) is adapted and used here.

Comment 5: it sounds like the exhumed kerogen was all delivered to the oceans and got buried in marine sediments during the interested timescale of deglaciation (e.g. Figure 1) – was this true?

Sediment residence time in floodplains and sedimentary basins could reach tens of thousands of years – meaning some of the kerogen might not be delivered to the oceans during the deglaciation. Then, would the conditions in floodplains and sedimentary basins also influence kerogen carbon reburial efficiency? can add relevant discussions.

I totally agree. Intermediate traps of kerogen on land certainly add a layer of complexity that has gone largely unaddressed in studies to date. I have added a short discussion on this: *In tandem with this, quantification is needed for (temporary) kerogen reburial in subaerial and subaquatic terrestrial systems (e.g., moraines, lakes) on global and regional scales (e.g., Meybeck, 1993; Vonk et al., 2016; Blattmann et al., 2019b; Fox et al., 2020).*

Title: can be more focused and straightforward – sth like 'oxidation of kerogen contributed to CO2 rise at glacial terminations'

The author disagrees. Making a strong statement in the title like this would be too strong for a presentation of ideas and perspectives. A dynamic exogenous kerogen cycle was also suggested for other events in Earth's history by other studies, such as during the PETM and this literature is unified in this contribution.

L25: please clarify how the 150 PgC/kyr was determined? uncertainties?

This is from the review by Hedges and Oades (1997). While this estimate will likely be revised in the future with improved budgeting of the fluxes, changes to this number produce no changes to the thoughts presented here. Please see response to reviewer 1 for further discussion on this value and its negligible sensitivity to the end result of this work.

L30-35: should also introduce major thoughts of the causes of glacial terminations

Thank you for this suggestion. I have added a sentence at the beginning of section "2 Carbon isotopes and contradictions?" to get the reader on the same page.

L40-45: this carbon cycle framework is very incomplete – at least should put in silicate weathering, see more in Berner et al. (1983)

Thank you for this constructive suggestion. I have reorganized section "3 Kerogen and glaciers" by adding an introductory section that does carbonate and silicate weathering justice. I totally agree, this was treated in a very peripheral manner and now it is given proper context and helps guide the reader in a balanced way. Please see lines 66-78.

L100: Equation 2 is unclear...explain what Zeng (2003), Simmons et al. (2006), and Horan et al. (2017) have done? Where were those studies conducted? What did they find?

The equations have been revised so that variables are used. I believe these references are sufficiently described and flow of ideas is maintained by keeping this part of the work straight and to the point. Please see lines 105-120.

L125-130: how much did the 14C composition of the then atmosphere-ocean carbon reservoir change? Any comment on 13C?

All of this is reported graphically and referenced with Fig. 3.

L230-235: could expand a bit and discuss some existing mechanisms – their pros and cons?

Thank you. I have expanded briefly on this and refer to reviews that provide the space to discuss pros and cons in detail: Various mechanisms have been proposed to explain CO2 increases at glacial interglacial transitions including for example the solubility pump hypothesis, iron fertilization hypothesis, ocean circulation hypotheses and many more (see hypotheses and reviews by Martin, 1990; Broecker and Peng, 1993; Kohfeld and Ridgwell, 2009; Rapp, 2019).

L500 – Table 1: how did the laboratory experiment-based results translate to a flux of unit area? for example, bituminous coal and oil sands – how to convert the reaction kinetics results of several samples to fluxes over certain areas of landscapes?

Thank you. In response to this comment, I have made an online supplemental which details the conversion of the data from the references contained in table 1.

L525: Figure 4 can be improved by displaying topography and lithology maps

The author disagrees. Overlaying additional maps make the figure too busy. Refinement comes later as the subject of future work. The map already contains geographical information, major geological units, and two isochrons for the extent of the Laurentide Ice Sheet for the time bracket of interest. Adding more information will make the reader lose the core message. References are clearly given (with a new addition) and the interested reader can delve deeper into lithological maps, etc. to explore for themselves using dedicated maps. The focus needs to be on the temporal and megascale dimensions of the phenomenon: namely that CO2 rises relentlessly throughout the ice sheet's retreat across the sedimentary units of western Canada and reaches its inflection point shortly after the ice sheet recedes into the Canadian Shield. This is the important feature and the core of the hypothesis as manifested in the known history of the deglaciation of North America.

---

## Referee Report (RR1)

**Review of « Ideas and perspectives: Emerging contours of a dynamic exogenous kerogen cycle » by Thomas Blattmann**

I am happy to see that, finally, this new version of the manuscript attempts to discuss the question of the carbon isotopic budget as a fundamental constraint on the origin of the deglacial atmospheric carbon increase. But still, the author seems to cherry pick only some oceanic data as a way to stick to his original hypothesis and therefore does not provide a fair account of the litterature on this topic. As a result, I feel the author tries to blur the marine isotopic evidence in order to make his point that kerogens contributed significantly to the pCO2 glacial-interglacial increase. Overall, I believe this is damaging to the paper, since it does not provide a fair account of the available litterature on this topic and a fair account of the actual numbers.

**In the response :**

« generalizing the positive Last Glacial Maximum to Holocene  $\delta^{13}C$  shift to the global oceans is imprecise... »

In the revised paper :

« the global deglacial increase in carbon isotopes shows a notable exception: For much of the North Atlantic, the Holocene stable carbon isotope values of DIC are lighter than those of the Last Glacial Maximum ... This is notable because the northernmost Atlantic is the locus of major downwelling driving global thermohaline circulation ... »

The question is not to « generalize » the isotopic shift to the world ocean, but simply to compute the **net global budget**. From the above Fig. (from Peterson et al 2020) the author does « cherry-pick » the North Atlantic intermediate waters (the top part of the red curves) as an example of oceanic 13C data that heavier during the last glacial. Without doing any complex computation, my conclusion from this Figure is that most of the Ocean (and in particular the heavy players like the Pacific) are lighter during LGM. Since the DIC in the ocean accounts for about 95% of the Earth surface carbon, since most curves on this figure are negative, since doing carbon (or any) budget implies accounting first for the largest reservoirs, I conclude that the **global carbon** signature was negative, therefore the contribution of light carbon (living organic matter, permafrosts or kerogens) is globally to stock more carbon during the Holocene than during the LGM. Again, this is well known for many decades in the carbon community and it stands indeed as a major contraint. It therefore « does not help » to solve the pCO2

increase, but on the contrary raises the burden for the oceanic contribution. Peterson et al (2020) conclude their paper with a revised estimate of this global budget, on the Figure below (the stars with error bars).

**Figure 5.** Comparison of our deep-ocean and whole-ocean estimates with previous studies. Marine estimates are denoted by blue squares, our estimates are blue stars; terrestrial carbon storage change estimates for these studies are calculated using equations (1) and (2), as indicated in Table 1. Model-based estimates are red diamonds, and pollen-based and vegetation-based reconstructions are black circles. The error bars on our estimates for other studies are omitted for clarity. S77 = [*Shackleton*, 1977]; C88 = [*Curry et al.*, 1988]; D88 = [*Duplessy et al.*, 1988]; B92 = [*Boyle*, 1992]; C95 = [*Crowley*, 1995]; AF98 = [*Adams and Faure*, 1998]; MLS99 = [*Matsumoto and Lynch-Stieglitz*, 1999]; K10 = [*Köhler et al.*, 2010]; P11 = [*Prentice et al.*, 2011]; C11 = [*Ciais et al.*, 2011].

It is interesting to note that ALL estimations since the very first one (Shackleton 1977) agree that the **MEAN** ocean 13C signal was lighter during the LGM, and therefore that the (terrestrial) light carbon stocks (living organic matter, permafrosts or kerogens) were therefore smaller during LGM by several hundreds of GtC. This appears to me quite a strong and robust consensus on this question, and it seems to me not fair to avoid this piece of evidence by downplaying it. Of course, if the deglacial terrestrial vegetation regrowth is very large, this may allow for a significant release of permafrosts or kerogens : the isotopic constraint applies only to the overall budget.

**In the paper :**

« In contrast to DIC of the oceans, atmospheric carbon isotope composition of CO2 directly measured from ice core recovered CO2 reflects a well-mixed, global signal. »

Indeed, but it only accounts for 1 or 2% of the Earth surface carbon (about 600 GtC compared to 40000 GtC): the isotopic signal is interesting for the dynamics of the deglaciation, in particular the timing of the different contibutions (vegetation, permafrost, ocean, ...) since it stands « at the center » of these exchanges. But it is certainly not very relevant for the overall glacial-interglacial budget.

**In the paper :**

« Reconstructed stable carbon isotope composition of DIC stems primarily from foraminifera which may also include bias from vital effects (e.g., Erez, 1978; Spero et al., 1997; Lea et al., 1999; see also Schmittner et al., 2017). Unlike the global, nearly unison rhythm of the glacial-interglacial marine oxygen isotope record, the global deglacial increase in carbon isotopes shows a notable exception »

There are also many notable exceptions in the oxygen isotopes... as well as many unconstrained vital effects. Still, the carbon isotopes measured in modern foraminifera follows closely the carbon isotopes measured in modern seawater, and they have been calibrated and used for almost 50 years as THE main tracer of carbon in the ocean in

paleoceanography. I therefore do not agree with the author's sentence, whose purpose seems only to avoid discussing seriously the isotopic budget problem.

**In the response :**

« in summary, the modeling work by Ciais et al. (2012) and Crichton et al. (2016) suggest that the observed  $\delta$ 13C patterns in atmosphere and ocean are compatible with kerogen oxidation. »

Of course they are... WHEN accounting for the problem and accepting that (basically) MORE than 100% of the glacial-interglacial carbon came from the ocean, since the NET organic matter contribution is globally negative. The figures below (from Crichton et al. 2016), also cited by the author in his response, are very explicit on this point when discussing the role of permafrost.

---

## Author Response (AR2)

**Response to Associate Editor**

Associate Editor Decision: Reconsider after major revisions (18 May 2021) by Markus Kienast

Comments to the Author:

Dear Thomas,

Your idea continues to intrigue the referees. Despite favourable reviews by some colleagues, one of the main concerns/objections remain, the oceanic 13C signal. Together with the critical referee, I perceive this to be a key data constraint that you need to reconcile with your hypothesis. Thus, unless you provide a thorough discussion and reconciliation of your idea with the overall 13C constraints on the Earth's carbon budget, I am afraid I cannot accept your contribution for publication.

Sincerely,

Markus

**Dear Markus, Editor,**

Thank you very much for your review. The issue of reconciling atmospheric with marine  $\delta^{13}$ C arose with the emergence of increasingly detailed atmospheric  $\delta^{13}$ C records into the 2000s. These atmospheric  $\delta^{13}$ C records prompted a wave of hypotheses invoking a terrestrial organic matter source of carbon to the atmosphere (e.g., Bauska et al., PNAS, 2016; Tesi et al., Nat. Commun., 2016; Crichton et al., Nat. Geosci., 2016; Martens et al., Sci. Adv., 2020; Winterfeld et al., Nat. Commun., 2018; Lindgren et al., Nature, 2018; Köhler et al., Nat. Commun., 2014; Ciais et al., Nat. Geosci., 2012) with however only few addressing the overall  $\delta^{13}$ C constraints of Earth's carbon budget.

For the responses and revisions, I have assembled my thoughts on marine  $\delta^{13}C$  and its compatibility with the kerogen oxidation hypothesis leading to an improved discussion in section 2 "Carbon isotopes and contradictions?". I rebut a few of the comments made by the reviewer which includes clarifying the findings of Ciais et al. (2012). Using  $\delta^{13}$ C and additional constraints, Ciais et al. (2012) concluded that with a release of 700 PgC from an "inert" carbon terrestrial carbon pool, the growth of the terrestrial biosphere from glacial to interglacial was smaller than many studies previously surmised, i.e., only 300 PgC of increase rather than the 300-700 PgC increase reported by most other studies. As elaborated in the technical response to the reviewer, this is compatible with globally observed  $\delta^{13}$ C trends for the atmosphere and oceans. In the case of the latter, generalizing the positive Last Glacial Maximum to Holocene  $\delta^{13}$ C shift to the global oceans is imprecise: the North Atlantic, which is fed directly by deep water formation in the global ocean conveyer belt, exhibits a negative  $\delta^{13}$ C shift from Last Glacial Maximum to Holocene (e.g., Peterson et al., 2020; Broecker and McGee, 2013). The opposing directions of  $\delta^{13}$ C change in different parts of the ocean was modeled by Crichton et al. (2016) who demonstrate that these observations are compatible with the release of an isotopically light carbon source from land reflected in the  $\delta^{13}$ C trajectory of the North Atlantic. In summary, the modeling work by Ciais et al. (2012) and Crichton et al. (2016) suggest that the observed  $\delta^{13}$ C patterns in atmosphere and ocean are compatible with kerogen oxidation.

Glacial-interglacial cyclicity remains one of the great mysteries of our time. This *Ideas and Perspectives* article provides a concise overview of the state of knowledge on kerogen weathering that I hope will stimulate constructive discussion and research action in hitherto unexplored directions – in particular in its hypothesized connection with glacial-interglacial cyclicity. After two years of manuscript development driven by rigorous and constructive peer review as well as stimulating input by colleagues and readers online, the contribution has made another marked improvement with an outcome that I am eager to formally present to the readers of *Biogeosciences*.

Thank you very much for your editorial handling. I look forward to your response.

Sincerely,

Thomas 28.06.2021 Zurich

**Response to Report #2, Referee #4**

This manuscript deals with most comments from the reviewers. Still I am disappointed that it fails to address the question raised during its first submission concerning the overall 13C constraints on the Earth's carbon budget (https://bg.copernicus.org/preprints/bg-2019-273/bg-2019-273-RC2.pdf).

Indeed, a strong point against the idea that kerogens and more generally organic matter could have a decisive contribution in the glacial-interglacial CO2 problem is the well-established observation that the global ocean 13C was about 0.3‰ lower during glacial times and increased accordingly during the deglaciation. Since the 1970s, this is taken as a proof that the total amount of organic matter (« living or dead», forests or buried) was smaller during glacial times. In contrast, releasing carbon from kerogens to Earth's atmosphere and ocean, as suggested by the author, would lower oceanic 13C but not increase it. Looking at the atmospheric 13C is not quite relevant since the main carbon reservoir is the ocean. Atmospheric shows very interesting 13C variations (first a decrease then an increase) that are associated with the dynamics of carbon during the deglaciation, but it is certainly not constraining the overall glacial-interglacial carbon budget since it contains only about 600 GtC (the ocean about 38000 GtC): if the glacial-interglacial carbon change is due to low 13C carbon released during the deglaciation, then the ocean 13C should decrease. It does not.

The same is also true for carbon stored in permafrost areas, which were probably significantly larger during glacial times and thus released large amounts of carbon during deglaciation. The only way to account for such a release of « dead carbon » in the glacial-interglacial 13C problem is to consider a significantly reduction of the living biosphere on the continents, since forests represent the largest « living carbon » reservoir. This was discussed for instance in Ciais et al. (Nature Geosciences, 2011).

In other words, it is indeed quite possible that a significant release of kerogens and permafrost carbon occurred during the deglaciation, but this must be over-compensated in terms of forests reduction in order to have an overall negative contribution of organic matter to the carbon budget. Finding new sources of deglacial organic matter release does not help to solve the glacial-interglacial CO2 problem since the overall effect of organic matter (forests, permafrost or kerogens) must be negative.

Ciais et al. (2011) Large inert carbon pool in the terrestrial biosphere during the Last Glacial Maximum DOI: 10.1038/ngeo1324.

**Dear Reviewer,**

I agree, I inadequately addressed the 13C constraints on Earth's carbon budget. The longstanding issue of the overall constraints on the Earth's carbon budget over glacial-interglacial cycles is key to understanding our Earth System. Thanks to your input, I have improved the discussion in section 2 "*Carbon isotopes and contradictions?*" to highlight the 13C mystery – a mystery which, as I see it, begins with the primary datasets themselves, irrespective of the hypothesis one supports.

I begin with responding to and rebutting a few lines in the review: "Looking at the atmospheric 13C is not quite relevant since the main carbon reservoir is the ocean. Atmospheric shows very interesting 13C variations (first a decrease then an increase) that are associated with the dynamics of carbon during the

deglaciation, ..." (Referee #4). First of all, 13C of the atmosphere is relevant because it is a direct marker for change in atmospheric CO2, which is directly relevant for global climate (unlike marine dissolved inorganic carbon, which has no immediate influence on climate in terms of a greenhouse gas effect). However, regarding your second point, I fully agree: the atmosphere does show very interesting 13C variations, which are associated with the dynamics of carbon during deglaciation and this contribution hypothesizes kerogen oxidation to explain a part of these 13C variations. Regarding a third point, in discussing the conclusions from Ciais et al. (2012): "The only way to account for such a release of « dead carbon » in the glacial-interglacial 13C problem is to consider a significantly reduction of the living biosphere on the continents" (Referee #4). No, what Ciais et al. (2012) concluded was different. Ciais et al. (2012) concluded that with a release of 700 PgC from an inert carbon terrestrial carbon pool (which they ascribe to oxidation of organic carbon from permafrost), the growth of the terrestrial biosphere from glacial to interglacial was smaller than many studies previously surmised, i.e., only 300 PgC of increase rather than the 300-700 PgC increase reported by most other studies. This is possible as during the redistribution of carbon from inorganic to organic pools, isotope fractionation occurs with increased selectivity for lighter carbon at increasingly higher atmospheric CO2 concentrations (Farquhar et al., 1989; see also Broecker and McGee, 2013). Therefore, as supported by the study of Ciais et al. (2012), the idea of oxidation of an "inert" terrestrial organic carbon pool alongside the growth of the terrestrial biosphere as highlighted in Fig. 3 of this manuscript is compatible with the atmosphere's and ocean's carbon isotope trajectory.

Starting from the primary datasets, oceanic and atmospheric  $\delta^{13}$ C show different trends and signal characteristics. Atmospheric  $\delta^{13}$ C of CO2 directly measured from ice core recovered CO2 reflects a globally well-mixed signal which varies through time in a steadier manner than marine dissolved inorganic carbon (e.g., compare Schmitt et al., 2012 and Galaasen et al., 2020). Glacial-interglacial marine  $\delta^{13}$ C of dissolved inorganic carbon reconstructed from the foraminifera proxy, incorporating some bias and uncertainty (e.g., Spero et al., 1997; Lea et al., 1999; see also Fig. 7 in Schmittner et al., 2017), are spatially variable in their temporal patterns. In contrast to well-behaved global marine  $\delta^{18}$ O pattern, the increase in  $\delta^{13}C_{DIC}$  is not observed moving into interglacials in parts of the ocean (see Fig. 1 in this response showing global compilation). In fact, the opposite of the Shackleton generalization is observed for much of the North Atlantic, where the Holocene  $\delta^{13}$ C values are lighter than those of the Last Glacial Maximum (see Fig. 2 in response; e.g., Peterson et al., 2020; Broecker and McGee, 2013). This is notable because the northernmost Atlantic is the locus of major downwelling feeding global thermohaline circulation (de Carvalho Ferreira and Kerr, 2017). This negative shift observed in the N. Atlantic was modeled by Crichton et al. (2016) (see Fig. 3 in response) who using Ocean-Land-Atmosphere models show this is explainable by the marine uptake and subduction of light carbon released to the atmosphere by terrestrial organic matter oxidation (hypothesized as permafrost in their case). Similarly, Broecker and McGee (2013) also conclude that the LGM-Holocene shift in atmospheric  $\delta^{13}$ C towards heavier values is smaller than would otherwise be expected based on changes in 1) photosynthetic isotope fractionation, 2) air-sea exchange due to change in temperature, and 3) the upper ocean  $\delta^{13}$ C. One explanation towards closing this gap between expected and measured  $\delta^{13}$ C in the atmosphere would be to include a source of light carbon as suggested in this contribution (see Fig. 2 in Broecker and McGee 2013). Going a step further, using the input of a light carbon sourced from land in their Ocean-Land-Atmosphere models, Crichton et al. (2016) model the positive  $\delta^{13}$ C shift observed in the S. Atlantic demonstrating that contemporaneous positive and negative shifts, as observed in primary datasets from the southern and northern sectors of the Atlantic ocean, respectively, are compatible with the release of isotopically light carbon from land. In

summary, global atmospheric  $\delta^{13}$ C records and region-specific marine  $\delta^{13}$ C records are reconcilable with the release of CO2 via kerogen oxidation operating alongside classically considered processes as suggested by the results by Ciais et al. (2012), Crichton et al. (2016), and others. However, in the author's opinion, much more needs to be done in future work to constrain the glacial-interglacial carbon budget: for example, marine dissolved organic matter (similar in carbon amount to the atmosphere) remains poorly constrained beyond the Holocene (e.g., Wagner et al., 2020) and a global, holistic carbon budget would require that all carbon pools are accounted for.

With the emergence of atmospheric  $\delta^{13}$ C records into the 2000s, hypotheses invoking a terrestrial organic matter source of carbon to the atmosphere were proposed (e.g., Bauska et al., PNAS, 2016; Tesi et al., Nat. Commun., 2016; Crichton et al., Nat. Geosci., 2016; Martens et al., Sci. Adv., 2020; Winterfeld et al., Nat. Commun., 2018; Lindgren et al., Nature, 2018; Köhler et al., Nat. Commun., 2014; Ciais et al., Nat. Geosci., 2012). However, the spatiotemporal pattern of glacial retreat of the Laurentide Ice Sheet from kerogen-rich substrates and concurrent changes in atmospheric chemistry point to the intriguing possibility that kerogen oxidation played a significant role in the deglacial rise of CO2. The review component of this contribution lays out a roadmap for advancing our basic knowledge of kerogen cycling to work towards testing this hypothesis.

Sincerely,

Thomas Blattmann 28.06.2021 Zurich

Figure 1: Oxygen and carbon isotope variations over the past 800,000 years for different ocean sites globally (Bouttes et al., 2020). Oxygen isotope variations show a globally rhythmic pattern (left panel), while carbon isotope records show some arhythmic sedimentary sequences (right panel). North Atlantic sites display regionally distinct behavior compared to other oceans (right panel).

Figure 2: Left: Average Last Glacial Maximum-Holocene change in the carbon isotope composition of marine dissolved inorganic carbon averaged across different oceans and depths (Peterson et al., 2020); Right: Last Glacial Maximum-Holocene change in the carbon isotope composition of marine dissolved inorganic carbon for the North Atlantic with a negative shift observed at depths

---

## Author Response (AR3)

**Response to Report #1 from Referee #4 « Ideas and perspectives: Emerging contours of a dynamic exogenous kerogen cycle » by Thomas Blattmann**

I am happy to see that, finally, this new version of the manuscript attempts to discuss the question of the carbon isotopic budget as a fundamental constraint on the origin of the deglacial atmospheric carbon increase. But still, the author seems to cherry pick only some oceanic data as a way to stick to his original hypothesis and therefore does not provide a fair account of the litterature on this topic. As a result, I feel the author tries to blur the marine isotopic evidence in order to make his point that kerogens contributed significantly to the pCO2 glacial-interglacial increase. Overall, I believe this is damaging to the paper, since it does not provide a fair account of the available litterature on this topic and a fair account of the actual numbers.

Dear Reviewer,

Thank you for your review and keeping up the pressure to motivate me for another round of improvement. As a result, I feel the contribution has reached a very high level. Now included is an example of a carbon budget that illustrates that the hypothesis of kerogen oxidation is both plausible and compatible with the carbon isotope trajectories of both the atmosphere and marine DIC.

Along with improvements to the text, I am convinced that this contribution will promote constructive discussion between interdisciplinary communities and will motivate new research surrounding glacial-interglacial cycles as well as kerogen oxidation. We have much to learn about both.

Sincerely,

Thomas Blattmann

19.10.2021 Zurich

In the response :

« generalizing the positive Last Glacial Maximum to Holocene  $\delta13C$  shift to the global oceans is imprecise... »

In the revised paper :

« the global deglacial increase in carbon isotopes shows a notable exception: For much of the North Atlantic, the Holocene stable carbon isotope values of DIC are lighter than those of the Last Glacial Maximum ... This is notable because the northernmost Atlantic is the locus of major downwelling driving global thermohaline circulation ... »

The question is not to « generalize » the isotopic shift to the world ocean, but simply to compute the net global budget. From the above Fig. (from Peterson et al 2020) the author does « cherry-pick » the North Atlantic intermediate waters (the top part of the red curves) as an example of oceanic 13C data that heavier during the last glacial.Without doing any complex computation, my conclusion from this Figure is that most of the Ocean (and in particular the heavy players like the Pacific) are lighter during LGM. Since the DIC in the ocean accounts for about 95% of the Earth surface carbon, since most curves on this figure are negative, since doing carbon (or any) budget implies accounting first for the largest reservoirs, I

conclude that the global carbon signature was negative, therefore the contribution of light carbon (living organic matter, permafrosts or kerogens) is globally to stock more carbon during the Holocene than during the LGM. Again, this is well known for many decades in the carbon community and it stands indeed as a major contraint. It therefore « does not help » to solve the pCO2 increase, but on the contrary raises the burden for the oceanic contribution. Peterson et al (2020) conclude their paper with a revised estimate of this global budget, on the Figure below (the stars with error bars).

It is interesting to note that ALL estimations since the very first one (Shackleton 1977) agree that the MEAN ocean 13C signal was lighter during the LGM, and therefore that the (terrestrial) light carbon stocks (living organic matter, permafrosts or kerogens) were therefore smaller during LGM by several hundreds of GtC. This appears to me quite a strong and robust consensus on this question, and it seems to me not fair to avoid this piece of evidence by downplaying it. Of course, if the deglacial terrestrial vegetation regrowth is very large, this may allow for a significant release of permafrosts or kerogens : the isotopic constraint applies only to the overall budget.

The revised main text now includes a carbon budget. This carbon budget is pegged to multiple parameters including the size and carbon isotope composition of atmospheric CO2, marine DIC, marine DOC, and terrestrial biosphere carbon pools in the Holocene, transition phase, and the Last Glacial Maximum. Additional constraints are set by assuming that marine DOC remained constant in size, the ratio of C3 to C4 biospheric mass is estimated at 4:1 during the Holocene according to areal distribution and productivity constraints provided by Still et al. (2003). Furthermore, stable carbon isotope constraints for marine DIC and atmospheric  $CO_2$  are pegged to paleorecord values and marine DOC,  $C_4$ , and  $C_3$  values were fixed according to literature values. Additionally, the kerogen oxidation component was set at 600 PgC which previous studies (e.g., Zeng, 2003) proposed based on models using radiocarbon and other constraints while the carbon isotope composition of kerogen-derived  $CO_2$  was fixed at -25‰. For this latter value, observational or experimental values for the relationship between bulk kerogen and kerogen-derived CO2 are lacking in the literature; however, given the many uncertainties this represents a reasonable assumption that still illustrates the main point of the proof-of-concept carbon budget. The budget strictly requires 1) carbon mass balance and 2) carbon isotope mass balance. With these geochemical and mathematical constraints, an array of solutions is possible, however, the budget presents a plausible set of numbers which suggests a growth in the terrestrial biosphere on the order of 1000 PgC – coincidentally in the ballpark of Shackleton's (1977) estimate. This estimate aligns with palaeoecological studies suggesting growths of this size – larger than most geochemical estimates based solely on the  $DI^{13}C$  shift. An important degree of freedom is the ratio of C4 to C3 vegetation, which is needed to maintain carbon isotope mass balance.  $C_4$ - $C_3$  shifts are expected given the change in vegetation across glacial-interglacial transitions; the scenario suggested in Table 1 shows an increasing proportion of C3 vegetation aligning with such expectations. While each parameter contains uncertainty (e.g.,  $+0.34\pm0.19$ % 2- $\sigma$  increase for global marine DIC, Peterson et al., 2014), many parameters are fair game for debate, and the parameter space/sensitivity is explorable in many directions, the key point immediately relevant for this contribution is plausibly illustrated: kerogen oxidation is compatible with the global carbon isotope mass budget – both with trends in atmospheric CO2 and marine DIC and a regrowing biosphere.

In the paper :

« In contrast to DIC of the oceans, atmospheric carbon isotope composition of CO2 directly measured from ice core recovered CO2 reflects a well-mixed, global signal. »

Indeed, but it only accounts for 1 or 2% of the Earth surface carbon (about 600 GtC compared to 40000 GtC) : the isotopic signal is interesting for the dynamics of the deglaciation, in particular the timing of the different contibutions (vegetation, permafrost, ocean, ...) since it stands « at the center » of these exchanges. But it is certainly not very relevant for the overall glacial-interglacial budget.

In the paper :

« Reconstructed stable carbon isotope composition of DIC stems primarily from foraminifera which may also include bias from vital effects (e.g., Erez, 1978; Spero et al., 1997; Lea et al., 1999; see also Schmittner et al., 2017). Unlike the global, nearly unison rhythm of the glacial-interglacial marine oxygen isotope record, the global deglacial increase in carbon isotopes shows a notable exception »

There are also many notable exceptions in the oxygen isotopes... as well as many unconstrained vital effects. Still, the carbon isotopes measured in modern foraminifera follows closely the carbon isotopes measured in modern seawater, and they have been calibrated and used for almost 50 years as THE main tracer of carbon in the ocean in paleoceanography. I therefore do not agree with the author's sentence, whose purpose seems only to avoid discussing seriously the isotopic budget problem.

« in summary, the modeling work by Ciais et al. (2012) and Crichton et al. (2016) suggest that the observed  $\delta$ 13C patterns in atmosphere and ocean are compatible with kerogen oxidation. »

Of course they are... WHEN accounting for the problem and accepting that (basically) MORE than 100% of the glacial-interglacial carbon came from the ocean, since the NET organic matter contribution is globally negative. The figures below (from Crichton et al. 2016), also cited by the author in his response, are very explicit on this point when discussing the role of permafrost.

The red curve corresponds to the « ocean-only » (including vegetation changes) contribution, which explains (more that) entirely the pCO2 rise as well as the (South Atlantic) oceanic 13C signal, due to a net increase in terrestrial organic carbon stocks linked to vegetation regrowth. This of course may leave some room for a permafrost (or a kerogen) contribution that may help explain the atmospheric 13C signal as shown in the Crichton paper, to the extent that it is smaller that the vegetation regrowth (since the net organic carbon contribution must remain negative).

To conclude, I want to stress that I have no objection against the author's hypothesis that kerogens may have some role in the deglaciation. But his paper would be much more interesting and valuable if it would present an unbiased view of the current knowledge on the glacial-interglacial carbon problem.

With the addition of a plausible carbon budget showing the compatibility of the presented hypothesis with marine DI13C, a key Earth system parameter, and a refocusing of the text the above concerns are addressed. However, unlike a review, this contribution is an "Ideas and perspectives" article. A perspective is always biased and was welcomed by another reviewer who got a kick out of reading the article – the first time for me to get this kind of feedback. However, I agree with your critique. As a result, I have rewritten the section "2 Carbon isotopes and contradictions?" clearly separating the literature review component from the perspective component. I think through this improvement the reader will get a fair

perspective and together with the "6 Synthesis and outlook" section will have a blend of literature references for further reading on the subject while at the same time maintaining a streamlined, fast-paced reading experience presenting perspectives and ideas on kerogen oxidation and its hypothesized connection with glacial-interglacial cycles.

Thanks to your review, I am convinced that the readers will receive a balanced perspective highlighting the caveats and that this contribution will stimulate new thinking towards testing this and other hypotheses surrounding biogeosciences, atmospheric chemistry, and glacial-interglacial cycles.

---

## Author Response (AR4)

**Response to Anonymous Referee #4**

The author has finally done what I was asking for: a "back-of-the-envelope" carbon budget (shown in table 1) that explicitly explains how his hypothesis can be reconciled with the isotopic data. Of course these numbers could be discussed lengthily. In particular, the author's hypothesis is requesting a very strong shift in C3/C4 plants that more than overcompensate the oceanic 13C changes, something that I have difficulties to buy. Still, the whole discussion being presented as an hypothesis, I think this paper is now acceptable.

The paper is also requesting that the glacial ocean carbon content was significantly larger (+520 GtC) and the terrestrial carbon significantly smaller (-920 GtC), which are both indeed key to explain lower glacial CO2 and oceanic 13C changes. I am therefore not entirely satisfied with sentences like "This work enunciates the possibility of kerogen oxidation as a major driver of atmospheric CO2 increase in the wake of glacial episodes" (abstract) since a large part (and in fact the most critical parts) of the carbon changes shown on table 1 are not linked at all to kerogens...

Anyway, as explained in the author's response: "this contribution is an "Ideas and perspectives" article. A perspective is always biased...".

I might agree with that to some extent, but I still would prefer a more balanced discussion.

Dear Reviewer,

Thank you very much for your feedback. I agree with your point and have therefor set emphasis on these other major processes in key parts of the text. Now, in the abstract, explicit reference to "… major oceanic degassing and biospheric regrowth…" is included and in one of the concluding paragraphs stronger wording is used to emphasize "…Earth system constraints such as the carbon isotope record dictate that other major processes must have acted." Also, the numerical output of the back-of-the-envelope calculation is now explicitly expressed in words in section 2, underscoring the necessity of multiple major sources and sinks working in tandem. Additionally, the abstract now includes explicit reference to the C3-C4 shift, which is certainly an interesting corollary from the isotope mass balance. Furthermore, the caption in figure 3 was edited to emphasize the major contributions of biospheric regrowth and ocean-air gas exchange to global biogeochemical cycles over the glacial-interglacial transition. Overall, the abstract, conclusions, figure caption of Fig 3, and the beginning as well as the end of discussion in section 2 provide high-visibility points emphasizing the necessity of multiple major carbon cycle drivers in the context of the kerogen oxidation hypothesis.

Thanks to your thorough and persistent feedback, this contribution has reached a level far higher than I could have imagined. This contribution together with the discussions resulting from this peer review process I am convinced will make for a stimulating resource of Ideas and Perspectives.

Sincerely,

Thomas Blattmann

06.12.2021 Zurich